# Life cycle net energy assessment of sustainable $H_2$ production and hydrogenation of chemicals in a coupled photoelectrochemical device

Xinyi Zhang[1,2], Michael Schwarze[2], Reinhard Schomäcker [2], Roel van de Krol [1,2] & Fatwa F. Abdi [1]✉

Green hydrogen has been identified as a critical enabler in the global transition to sustainable energy and decarbonized society, but it is still not economically competitive compared to fossil-fuel-based hydrogen. To overcome this limitation, we propose to couple photoelectrochemical (PEC) water splitting with the hydrogenation of chemicals. Here, we evaluate the potential of co-producing hydrogen and methyl succinic acid (MSA) by coupling the hydrogenation of itaconic acid (IA) inside a PEC water splitting device. A negative net energy balance is predicted to be achieved when the device generates only hydrogen, but energy breakeven can already be achieved when a small ratio (~2%) of the generated hydrogen is used in situ for IA-to-MSA conversion. Moreover, the simulated coupled device produces MSA with much lower cumulative energy demand than conventional hydrogenation. Overall, the coupled hydrogenation concept offers an attractive approach to increase the viability of PEC water splitting while at the same time decarbonizing valuable chemical production.

A transition from fossil to renewable energy is urgently needed to alleviate the climate problem caused by greenhouse gases (GHG) emitted from fossil-fuel-based energy generation. However, many renewable energy generation technologies, such as wind turbines and photovoltaic panels, rely heavily on intermittent sources; the supply of energy, therefore, has an unavoidable mismatch with the global energy demand. To overcome this limitation, energy needs to be stored. One promising option is to utilize sunlight to produce green hydrogen via solar water splitting. The hydrogen can be re-used to generate energy when the demand is high, either for stationary applications, as a mobile transportation fuel, or as a feedstock for various chemical transformations. Over the past decade, research on solar water splitting devices has achieved significant progress in terms of the demonstrated solar-to-hydrogen (STH) efficiencies. The highest efficiencies (up to

30%[1]) have been demonstrated using an indirect approach of combining photovoltaic (PV) cells and electrolyzers. Efficiency values exceeding 10% have been reported for direct solar water splitting with an integrated photoelectrochemical (PEC) device that combines both light absorption and electrochemical conversion functionalities within a single unit[2,3]. Such a configuration offers potential advantages over the indirect approach in terms of thermal coupling to improve the electrochemical reaction kinetics as well as the possibility to use cheaper and more abundant catalysts due to the 10–100× lower current densities[4].

Despite the impressive progress that has been made in this field, several techno-economic analyses (TEA) and net energy assessments (NEA) have indicated that the PEC approach is still not energetically and economically competitive for large-scale implementation.

[1]Institute for Solar Fuels, Helmholtz-Zentrum Berlin für Materialien und Energie GmbH, Hahn-Meitner-Platz 1, 14109 Berlin, Germany. [2]Technische Universität Berlin, Department of Chemistry, Straße des 17, Juni 124, 10623 Berlin, Germany. ✉e-mail: fatwa.abdi@helmholtz-berlin.de

The levelized cost of hydrogen (LCOH) produced from PEC systems has been estimated to be -10 USD/kg H₂, which is about an order of magnitude higher than that from steam methane reforming (-1.5 USD/kg H₂)[5]. In terms of the energy demand, a large-scale PEC facility requires up to 214 MJ/kg H₂[6], which is >20 times higher than the energy required by coupling wind turbines and electrolyzers (9.1 MJ/kg H₂[7]) and exceeds the energy content of the hydrogen itself (120 MJ/kg H₂, based on the lower heating value). A potentially attractive solution to increase the competitiveness of PEC systems is by coupling hydrogen production with the synthesis of valuable chemicals in a single reactor[8,9]. For example, instead of oxygen evolution, alternative oxidation reactions, such as the oxidation of 5-hydroxymethylfuran-2-carbaldehyde or the oxidation of sugars and lignocellulose, have been investigated[10]. Alternatively, a hydrogenation catalyst (homogeneous or heterogeneous) can be introduced to the catholyte, so that H₂ generated at the cathode can be partially used in situ to hydrogenate, e.g., biomass-derived feedstocks into valuable chemicals. This co-generation approach offers a route toward achieving an LCOH that is competitive with the current market price of hydrogen and increases the overall economic feasibility of PEC technology. The net energy balance of such systems, however, has not yet been studied in depth.

In this study, the coupling of homogeneously catalyzed hydrogenation of itaconic acid (IA) to methyl succinic acid (MSA) with hydrogen production inside a PEC water splitting device is evaluated. IA has been identified by the US Department of Energy as one of the 12 building blocks that possess the potential to be transformed subsequently into several high-value bio-based chemicals or materials[11]. MSA is a valuable chemical compound (with an estimated global market size of up to -15,000 $t$—see Supplementary Note 1), whose derivatives are ubiquitously used as solvents in cosmetics[12], polymer synthesis[13], binders in powder coatings[14], and organic synthesis, especially for pharmaceutical synthesis[15,16]. Hydrogenation of IA to produce MSA has been reported using hydrogenation catalysts in a conventional hydrogenation reactor at 25–150 °C and 1–140 bar H₂[17–20]. The feasibility of our proposed coupled approach is investigated by conducting a net energy balance assessment under various parametric scenarios and comparing the results with the benchmark values for conventional MSA production. Using several NEA metrics, we quantify the benefits of the coupled approach for the energy-generating performance of a PEC water splitting device.

## Results

### Investigated coupled PEC device

The assessed PEC device in this study is based on a tandem configuration consisting of a BiVO₄-based photoanode as the top absorber and a silicon heterojunction (SHJ) PV cell as the bottom absorber, as reported in the literature[21]. A modest STH efficiency ($\eta_{STH}$) of 5.5% has been reported using this tandem device configuration, and devices of up to 50 cm² have been demonstrated. A summary of the device components, their corresponding parameters, and the data source is shown in Table S1. Fig. 1a shows the schematic configuration of the PEC device. For simplicity, it is assumed that the Pt layer deposited on the

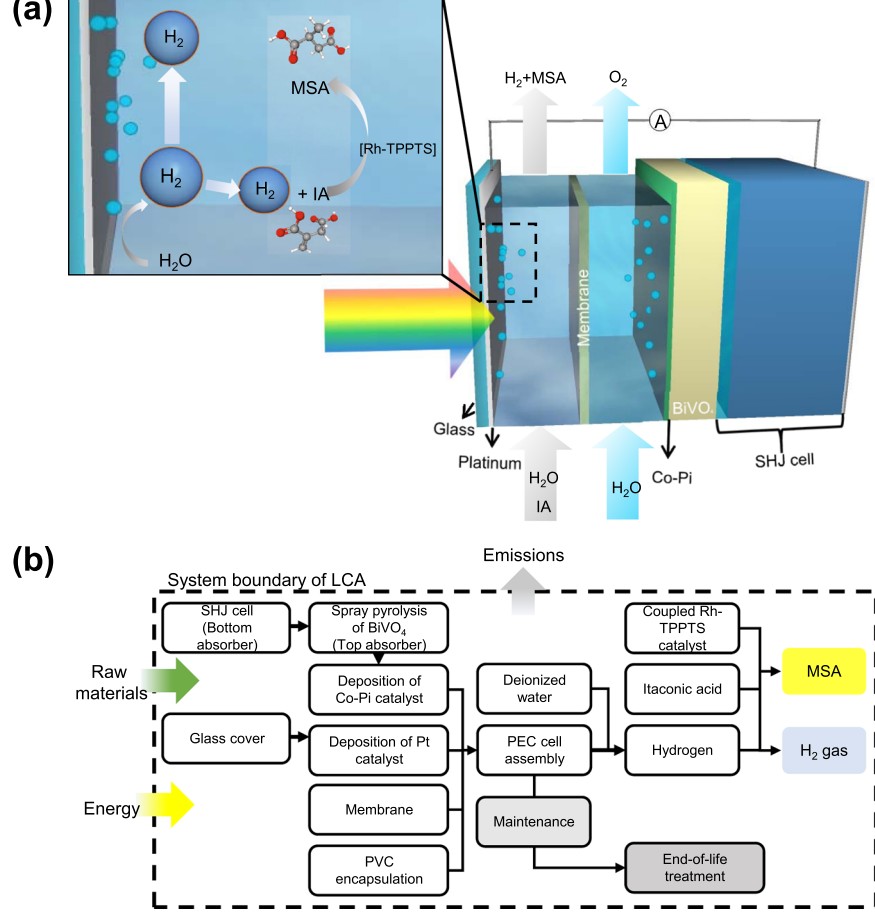

**Fig. 1 | Considerations used in the life cycle net energy assessment in this study.** **a** Schematic drawing of the coupled photoelectrochemical device considered in this study, in which hydrogen production and hydrogenation of itaconic acid (IA) to methyl succinic acid (MSA) occur in the catholyte chamber. **b** Simplified process flow diagram of the coupled PEC device for hydrogen production and hydrogenation of IA to MSA, starting from raw material extraction to the end-of-life treatment. The dashed line indicates the cradle-to-gate system boundary.

solar-grade glass is thin enough to transmit the sunlight; no optical loss is therefore considered in this study. Although this assumption is not accurate from the device operation point of view, and other geometrical architectures would likely be implemented, this factor would not significantly affect the overall analysis in this NEA study. A homogenous rhodium (Rh) trisodium 3,3′,3″-phosphanetriyltri(benzene-1-sulfonate) (TPPTS) catalyst complex is dissolved in the catholyte for the hydrogenation of IA to MSA. This specific catalyst is considered here since it has been reported to operate at room temperature and atmospheric pressure[22], which is the condition commonly used in PEC $H_2$ production. Fig. 1b shows the system boundary considered in this study, which covers the materials and fabrication processes of the device-scale PEC components for its cradle-to-gate life cycle. Periodic maintenance and end-of-life treatment are included in the preliminary system-level assessment. For maintenance, it is assumed that 10% of the device components are replaced annually[6]. As for the end-of-life treatment, National Energy Technology Laboratory (NETL) published a summary of the method used in their series of energy system life cycle assessments, which assumes that decommissioning requires 10% of the capital energy used for the initial construction of the system[23]. Further description of each component of the PEC device, the equations and assumptions used in our net energy assessment to generate several important NEA metrics (e.g., cumulative energy demand, CED; normalized net energy balance, NNEB; energy return on energy investment, EROEI; energy payback time, EPT) are provided in the Methods section. Uncertainty analyses were also performed considering three different scenarios (lower, base, and higher cases) at the device-level as well as the preliminary system-level assessments, as described in the Supplementary Notes 2 and 3.

## Cumulative energy demand of the coupled PEC device

Fig. 2a shows the distribution of cumulative energy demand for the fabrication of our coupled PEC device; the total energy required to produce 1 m$^2$ of the coupled PEC device is estimated to be 3834 MJ/m$^2$. Photoelectrodes material and fabrication consume ~70% of the total energy, and the SHJ bottom absorber is the most energy-intensive component in this category. The material required for the SHJ cell is the main contributor, while the fabrication process consumes 71% lower energy, as shown in Fig. 2b.

The finding that SHJ is the most energy-intensive component is interesting since one would not initially think that the silicon-based solar cell would be the main limitation. Our coupled PEC device consists of other elements that are less abundant and more expensive than Si. For example, the Rh used in the homogeneous catalyst is a rare element that has an extremely high energy demand of 683,000 MJ/kg[24]. However, since the amount of Rh used in the device is very small, its cumulative energy demand only takes up ~0.1% of the total energy. On the other hand, since the amount of Si used is relatively high (180 μm-thick), its cumulative energy demand ended up being the highest. For future optimization, the SHJ cell could be replaced with alternative absorbers that require less energy to produce. For instance, the cumulative energy demand of silicon microwire[25] and perovskite absorbers[26] have been reported to be more competitive at 661 MJ/m$^2$ and 779 MJ/m$^2$, respectively. Note, however, that they are not yet commercialized, and the energy demand is based on laboratory-scale data. Trade-offs between energy-saving and achievable efficiency typically exist, and they need to be further examined by simulations and/or experiments to justify the optimal component choice.

## Net energy balance and cumulative energy demand for 100%

The condition in which the PEC device produces only $H_2$ (i.e., no coupled hydrogenation reaction) is first considered. Table 1 shows the results from the net energy balance assessment for the PEC device under different scenarios, with $\eta_{STH}$ ranging from 3 to 10% and $t_{device}$

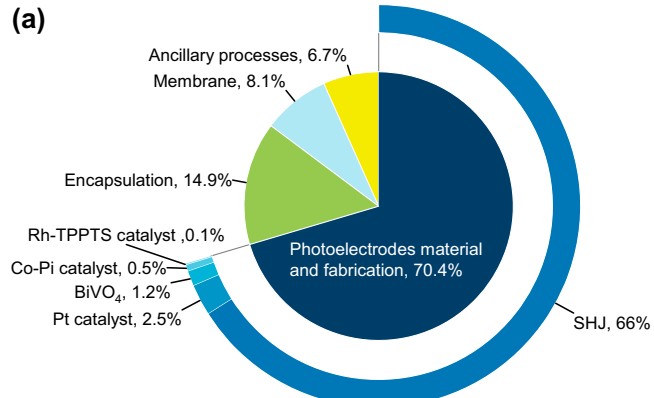

**(a)**

Ancillary processes, 6.7%
Membrane, 8.1%
Encapsulation, 14.9%
Rh-TPPTS catalyst, 0.1%
Co-Pi catalyst, 0.5%
BiVO$_4$, 1.2%
Pt catalyst, 2.5%
Photoelectrodes material and fabrication, 70.4%
SHJ, 66%

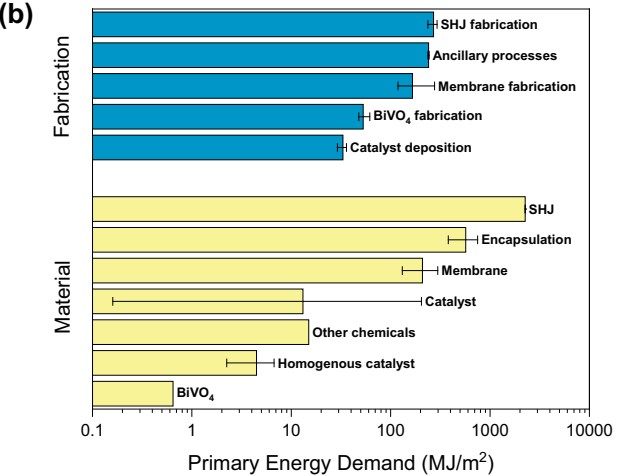

**(b)**

SHJ fabrication
Ancillary processes
Membrane fabrication
BiVO$_4$ fabrication
Catalyst deposition

SHJ
Encapsulation
Membrane
Catalyst
Other chemicals
Homogenous catalyst
BiVO$_4$

Primary Energy Demand (MJ/m$^2$)

**Fig. 2 | Cumulative energy demand of the coupled PEC device. a** Energy distribution pie-chart of our coupled PEC device indicating the contribution from the various components. Photoelectrodes material and fabrication consume the most energy (70.4%) and SHJ bottom absorber is the most energy-intensive component. **b** Logarithmic plot of the primary energy demand (MJ/m$^2$) of each component—divided into two categories: Material and Fabrication—of the coupled PEC device. Errors bars for cell fabrication and materials reflect the outcome from the uncertainty analysis.

**Table 1 | Net energy balance assessment results for PEC devices that produce only H$_2$ with various assumptions of STH efficiency ($\eta_{STH}$) and device longevity ($t_{device}$)**

| $\eta_{STH}$ | $t_{device}$ (years) | $CED_{kg\_H_2}$ (MJ/kg) | Net energy balance (MJ/kg) | Normalized Net Energy balance (MJ/m$^2$/year) |
|---|---|---|---|---|
| 3% | 5 | 680 | −560 | −631 |
| 3% | 10 | 340 | −220 | −248 |
| 3% | 28.3 | 120 | 0 | 0 |
| 3% | 30 | 113 | 7 | 7 |
| 5% | 5 | 408 | −288 | −541 |
| 5% | 10 | 204 | −84 | −158 |
| 5% | 17 | 120 | 0 | 0 |
| 5% | 30 | 68 | 52 | 98 |
| 10% | 5 | 204 | −84 | −316 |
| 10% | 8.5 | 120 | 0 | 0 |
| 10% | 10 | 102 | 18 | 67 |
| 10% | 30 | 34 | 86 | 323 |

$CED_{kg\_H_2}$ is the cumulative energy demand of the collected H$_2$, as defined in Eqs. (10) and (12).

ranging from 5 to 30 years. Under the base-case condition (i.e., $\eta_{STH}$ = 5%, $t_{device}$ = 10 years), the CED of $H_2$ is 204 MJ/kg. Because the LHV of $H_2$ is ~120 MJ/kg, the net energy balance is −84 MJ/kg. This translates to a negative normalized net energy balance of ca. −158 MJ/m²/year and the EROEI is 0.59 (see Eqs. (16) and (17) in the Methods section). This means that the energy output from the generated $H_2$ under the base-case conditions cannot compensate for the energy input required for the fabrication and operation of the device.

The energy payback time was also calculated for different $\eta_{STH}$ (see Eq. (18) in the Methods section). This indicates the minimum $t_{device}$ for the energy earned from the produced $H_2$ to be equal to the energy investment to build and operate the device, i.e., to obtain a zero net energy balance. The energy payback time obviously decreases with increasing $\eta_{STH}$: 28.3, 17, and 8.5 years would be needed for devices with $\eta_{STH}$ of 3%, 5%, and 10%, respectively. Only under the conditions in which $\eta_{STH}$ and/or $t_{device}$ is higher than these values would a positive net energy balance be achieved.

Table S5 lists the cumulative energy demand for generating $H_2$ using different production methods and compares it with our PEC device. $H_2$ produced by coupling wind electricity and electrolyzers is the most energy-efficient method with the CED of $H_2$ at only 9.1 MJ/kg[7]. The most common approach used in the industry is steam methane reforming (SMR) with the CED of $H_2$ at 183 MJ/kg $H_2$[27]. The production of hydrogen using electricity from a photovoltaic system to drive a water electrolysis unit has been reported to have an exergy efficiency of 0.64[28], which translates to a CED value of 187.5 MJ/kg $H_2$. The use of grid electricity (which is composed of a mix of different sources) and electrolyzers have been reported to yield a CED of $H_2$ in the same range as that using SMR[29]. The CED of $H_2$ generated from PEC water splitting

varies, based on the large variation of STH efficiency and device longevity. Values of 10 to 194 MJ/kg were reported in a study assessing a PEC device with a microwire Si absorber[25], while our study demonstrates values between 34 to 680 MJ/kg (Table 1). This comparison shows that only PEC devices with higher STH efficiency and/or device longevity would produce $H_2$ that is competitive vs. that produced using SMR or grid-electrolysis. However, it is unlikely that PEC water splitting will be energetically competitive vs. coupling wind electricity and electrolyzers.

### Net energy balance and cumulative energy demand for coupled production

We now turn our attention to the intended case for the coupled device, i.e., co-generation of hydrogen and hydrogenation of IA to MSA. It is expected that co-producing MSA can improve the energy performance of the PEC device, as MSA possesses a higher energy value than $H_2$. Only $H_2$-to-MSA conversion efficiency values of up to 60% are considered here; although higher conversion values are not theoretically impossible, this value (60%) represents the highest conversion efficiency already demonstrated in our preliminary experiments. The substitution method for energy allocation as discussed in the Methods section was applied. When $H_2$ is considered as the main product, the energy obtained from the MSA sub-product is subtracted from the total energy invested, as described by Eq. (7). The normalized net energy balance and the corresponding CED of $H_2$ can then be calculated. The results are shown in Table 2 under the base-case condition (i.e., $\eta_{STH}$ = 5%, $t_{device}$ = 10 years, see Fig. S2 for other $\eta_{STH}$ and $t_{device}$ values). As also already shown in Table 1, without the coupled reaction (i.e., zero $H_2$-to-MSA conversion efficiency), a negative net energy balance is obtained. Introducing the coupled hydrogenation reaction significantly increases the net energy balance to more favorable values. For example, converting 20% of the in situ generated $H_2$ to MSA already results in a much more positive normalized net energy balance value of 1655 MJ/m²/year. Consequently, the EROEI, in this case, is 3.8, which is much more favorable than the value of 0.59 when only $H_2$ production is considered.

Fig. 3a shows the colormap of the required $H_2$-to-MSA conversion efficiency to reach a zero net energy balance (i.e., energy breakeven) as a function of the STH efficiency and longevity of the PEC device. As expected, the higher the efficiency and the device longevity, the required $H_2$-to-MSA conversion efficiency for breakeven (i.e., an NNEB of 0 MJ/m²/yr) becomes lower. The significant benefit of the coupled reaction is clearly demonstrated by the following considerations. For a device with 5% STH efficiency and 10 years of longevity, energy

**Table 2 | Results of net energy balance assessment for co-producing $H_2$ and MSA under the base-case conditions of $\eta_{STH}$ = 5%, $t_{device}$ = 10 years with $H_2$-to-MSA conversion efficiency from 0% to 60%**

| STH efficiency | Longevity (years) | $H_2$-to-MSA conversion efficiency ($\mu$) | Normalized net energy balance (MJ/m²/yr) | CED of $H_2$ (MJ/kg) |
|---|---|---|---|---|
| 5% | 10 | 0% | −158 ± 63 | 204 ± 34 |
| 5% | 10 | 20% | 1655 ± 63 | −982 ± 42 |
| 5% | 10 | 40% | 3469 ± 63 | −2960 ± 56 |
| 5% | 10 | 60% | 5282 ± 63 | −6915 ± 84 |

The error values provided for the normalized net energy balance and the cumulative energy demand (CED) of $H_2$ were obtained from the uncertainty analysis.

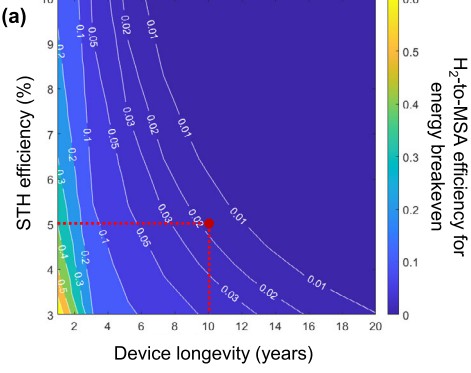
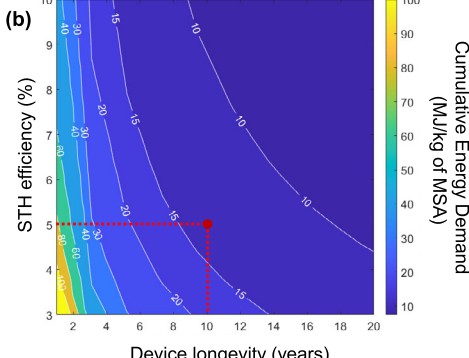

**Fig. 3 | $H_2$-to-MSA efficiency for energy breakeven and cumulative energy demand of MSA of the coupled device. a** Colormap of the required $H_2$-to-MSA conversion efficiency to achieve energy breakeven (i.e., zero net energy balance) for devices with various STH efficiencies and device longevities. Under the base-case scenario (red dashed lines), the required $H_2$-to-MSA conversion efficiency is ~2% for energy to breakeven (red circle). **b** The cumulative energy demand to produce 1 kg of MSA when the $H_2$-to-MSA conversion efficiency is 40% for various STH efficiencies and device longevities. Under the base-case scenario (red dashed lines), the cumulative energy demand is 13.3 MJ/kg MSA (red circle).

breakeven can already be achieved when $H_2$-to-MSA conversion efficiency is only ~2% (see red circle in Fig. 3a). Without the coupled reaction, a device with the same STH efficiency would only achieve energy breakeven when the device longevity is almost doubled to 17 years (see Table 1). In other words, the introduction of the coupled reaction relaxes the requirement on the efficiency and/or longevity of the device. If a $H_2$-to-MSA conversion of 20% can be achieved, the device longevity can be as short as 3 years even when the STH efficiency is only 3%.

When MSA is considered as the main product of the system and the same energy allocation method discussed previously is used, we can compare our MSA production with the benchmark MSA production by conventional hydrogenation. The cumulative energy demand to produce 1 kg of MSA at $H_2$-to-MSA conversion efficiency of 40% is shown in Fig. 3b for various STH efficiencies and device longevities. The CED of MSA decreases with increasing STH efficiency and longevity. Under the base-case condition (i.e., $\eta_{STH} = 5\%$, $t_{device} = 10$ years), the cumulative energy demand of MSA is 13.3 MJ/kg of MSA (see red circle in Fig. 3b) and would even be lower for higher $H_2$-to-MSA conversion efficiencies (see Fig. S3). In comparison, the benchmark MSA produced through the conventional approach requires 84 MJ/kg of MSA. The much lower energy consumption for PEC-produced MSA significantly increases its market competitiveness.

In order to evaluate the impact of parameters beyond the device level (e.g., balance-of-system, regional uncertainties, utility delivery, product handling, and separation), a preliminary system-level analysis was conducted considering a 100 m$^2$ (land area) coupled PEC system. The daily operation time of the system is assumed to be 12 hours, and the production of hydrogen gas and MSA occurs immediately upon exposure to sunlight[6,30,31]. Detailed discussion of additional parameters can be found in Supplementary Note 3, and the breakdown of the cumulative energy demand for the overall system is shown in Table S4. As expected, the inclusion of additional system-level inputs increases the required $H_2$-to-MSA conversion efficiency for energy breakeven. Under the base-case scenario, a 100 m$^2$ scale coupled PEC system requires 7.5% $H_2$-to-MSA conversion for energy breakeven. Although this is almost four-fold higher than the requirement from the device-level analysis, the 7.5% required $H_2$-to-MSA conversion is still well within the feasibility range considering that $H_2$-to-MSA conversion of up to 60% has been demonstrated in our preliminary experiments.

## Optimization of device components

Although the introduction of the coupled reaction into the PEC device already shows rather positive results, we can further optimize the energy performance by decreasing the primary energy demand of the PEC device. For example, certain alternative materials can be selected as promising substitutes for the device components. In this optimization, we focus again on the device-level analysis and ignore the system-level inputs. Two components are discussed here: the SHJ bottom absorber and the Nafion membrane as they consume the most energy in the device, i.e., 70% and 8% of the total energy demand, respectively. Table S6 lists various candidates for bottom absorbers and membranes, as well as their respective primary energy demands. Perovskite-perovskite (flexible substrate)[26] and polysulfone (PSF)[32,33] were selected for further assessment because of their low energy demands. Recently, Zhao et al. reported that the lifetime of perovskite absorbers is predicted to reach 5 years with a 17.4% efficiency[34]. Although a higher energy demand is expected due to more frequent replacement, perovskites still represent the most promising absorber alternative, especially since intense ongoing research efforts are likely to lead to further improvements in the material's stability. As a suitable membrane material, PSF has been proposed as a concrete alternative for Nafion with comparable performance[35]. According to the life cycle analysis by Yadav et al., the impact of PSF on the environment is lower than that of other membrane materials[36].

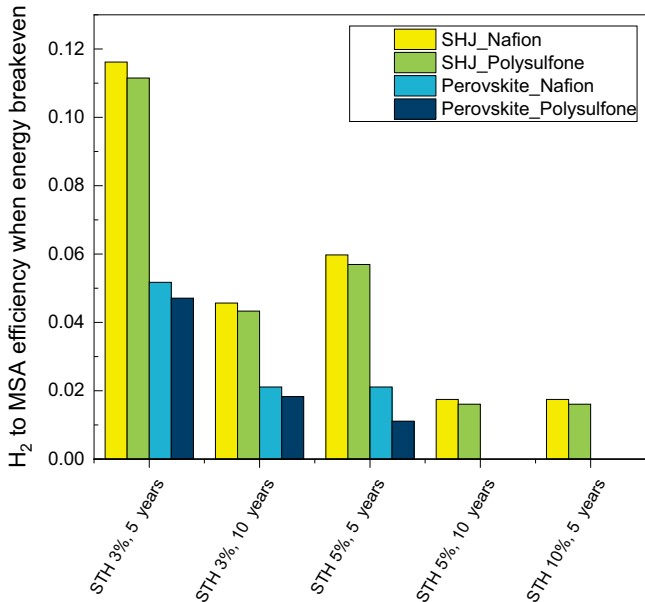

**Fig. 4 | Energetic impact of substituting components of the coupled PEC device.** Threshold $H_2$-to-MSA conversion efficiency for energy to breakeven in four cases of using different bottom absorber and membrane combinations in our coupled PEC device. No conversion efficiency is shown for case 3 (Perovskite_Nafion) and case 4 (Perovskite_Polysufone) when the STH is 10% and the longevity is 5 years since $H_2$ production itself is sufficient to pay back the energy investment, i.e., no $H_2$-to-MSA conversion is needed.

A total of four cases were compared based on the combination of the bottom absorber and membrane used: SHJ and Nafion (case 1), SHJ and polysulfone (case 2), perovskite and Nafion (case 3), and perovskite and polysulfone (case 4). Case 1 represents the base-case condition as discussed in section 3.1 with the CED of the PEC device at 3834 MJ/m$^2$. The cumulative energy demand of the PEC device for the other three cases is 3707 MJ/m$^2$, 2082 MJ/m$^2$, and 1955 MJ/m$^2$ respectively. Note that an additional 778.6 MJ/m$^2$ energy demand (CED of perovskite absorber) is included when the investigated longevity is >5 years since the perovskite lifetime is assumed to be 5 years. Under the base-case conditions (i.e., $\eta_{STH} = 5\%$, $t_{device} = 10$ years) and when the $H_2$-to-MSA conversion efficiency = 0.4, the normalized net energy balance can be increased by as much as 49% from case 1 to 4.

Fig. 4 shows the required $H_2$-to-MSA conversion efficiencies to achieve energy breakeven for the four different cases of devices under various STH efficiencies and device longevity. When the longevity is 5 years, the substitution of the bottom absorber from SHJ to perovskite reduces the required conversion efficiency by more than 50% (case 1 vs. 3) while the required conversion efficiency remains more or less unchanged when the membrane is changed from Nafion to polysulfone (case 1 vs. 2). This is not surprising since the reduction of the primary energy demand is much higher when replacing the bottom absorber instead of the membrane (see Table S6). However, when the longevity is 10 years, the replacement of perovskite-perovskite absorber increases the energy demand. As a result, the required conversion efficiency drops by 39.2% which is less than that in the 5-year scenarios.

When a perovskite is used as the bottom absorber (i.e., cases 3 and 4), $H_2$ production alone without any $H_2$-to-MSA conversion can already result in a positive net energy balance if the device performance is moderate ($\eta_{STH} = 10\%$ and $t_{device} = 5$ years). This simple analysis suggests that replacing the bottom absorber from SHJ with a perovskite would increase the competitiveness of our coupled PEC device even further.

## Discussion

In summary, we have investigated the energetic benefit of coupling photoelectrochemical (PEC) $H_2$ production with the hydrogenation of itaconic acid to methyl succinic acid. A coupled PEC device based on a $BiVO_4$ top absorber and silicon heterojunction (SHJ) bottom absorber was considered. The primary energy demand of the coupled PEC device is ~3800 MJ/m²; two-thirds of this value is attributed to the material use and fabrication of the SHJ bottom absorber. Alternative bottom absorbers that are less energy-intensive (e.g., a halide perovskite) would therefore decrease this primary energy demand. We then performed a net energy balance assessment and compared the case with and without the coupled hydrogenation reaction. Under the base-case condition (i.e., STH efficiency = 5%, device longevity = 10 years), producing only hydrogen yields a negative energy balance of *ca.* −160 MJ/m²/year. The introduction of the coupled hydrogenation of IA to MSA results in a more favorable energy balance. For example, a zero net energy balance (i.e., energy breakeven) can already be achieved when only 2% of the produced $H_2$ molecules are used to in situ hydrogenate IA to MSA. Under a more optimistic scenario with an $H_2$-to-MSA conversion efficiency of 40%, the net energy production is *ca.* 3500 MJ/m²/year, which translates to a cumulative energy demand of *ca.* 13.3 MJ/kg of MSA. This value is much lower than for MSA produced using conventional hydrogenation methods (i.e., ~90 MJ/kg MSA), which underlines the attractiveness of the coupled PEC approach. We note that this optimistic scenario is realistic since our preliminary experiments have demonstrated an $H_2$-to-MSA conversion efficiency of up to 60%.

We note that several limitations exist in our study that might introduce some degree of uncertainties to the presented results. First, the paucity of LCI data on some of the materials, certain fabrication processes, and the hydrogenation of IA to MSA required us to make assumptions based on either proxy materials or laboratory estimates. The accuracy of these assumptions is of course open to verification. In addition, since the investigated technology is still at a very low technology readiness level (TRL), many uncertainties regarding potential scale-up pathways are present. For example, likely, the most favorable fabrication method or process parameters found in the laboratory-scale experiment would be different than the ones for the industrial scale. Finally, many factors are still not yet considered in this device-level energy balance analysis, such as product separation, end-of-life treatment, maintenance, as well as techno-economic factors. Nevertheless, despite these limitations, our results clearly demonstrate the benefit of coupling hydrogen production and hydrogenation of chemicals, such as IA to MSA, inside a photoelectrochemical device.

Overall, our study provides solid evidence that coupling $H_2$ production and hydrogenation of chemicals in a PEC device is an attractive solution to increase the competitiveness of PEC water splitting devices. Future assessments regarding environmental impact and techno-economic performance should also be investigated to fully quantify the benefit of the coupled approach. Finally, other hydrogenation reactions possible to be performed at conditions relevant to that of PEC devices (e.g., near-atmospheric pressure, ambient temperature), such as levulinic acid to γ-valerolactonate[37] or acetone to isopropanol[38], need to be explored to determine the optimum reaction and/or identify the general applicability of the coupled device concept. Although some of these hydrogenation reactions may also be performed through electrochemical processes directly on suitable dark electrodes[39,40], the coupled concept proposed here introduces an important flexibility advantage in that the hydrogenation rate and the type of hydrogenation reactions performed can be controlled on demand using the same device without the need to change any components by simply changing the concentration and/or the types of feedstocks and homogenous catalysts.

## Methods

### Considerations used for the coupled PEC device

The total amount of hydrogen produced by the PEC device ($\Phi_{H_2\_produced}$ in kg/m²) is determined by the device performance and operational parameters:

$$\Phi_{H_2\_produced} = \frac{\Phi_{sunlight} \times \eta_{STH} \times 365 \times t_{device}}{LHV_{H_2}} \quad (1)$$

$\Phi_{sunlight}$ is the solar intensity in kWh/m²/day, $\eta_{STH}$ is the solar-to-hydrogen (STH) efficiency, $t_{device}$ is the device longevity or the service lifetime of the PEC device in years, and $LHV_{H_2}$ is the lower heating value of $H_2$ (33 kWh/kg or 120 MJ/kg). The average solar intensity in Germany of 3.4 kWh/m²/day was used in the base-case scenario in this study, while data from other locations in Germany were also chosen for the uncertainty analysis (see Supplementary Note 3)[41]. Since the PEC device is still at an early technology readiness level (TRL), significant uncertainties are present regarding future achievable $\eta_{STH}$ and $t_{device}$; a range of values was therefore considered for these parameters.

The in situ generated $H_2$ is then utilized to hydrogenate IA ($C_5H_6O_4$) to MSA ($C_5H_8O_4$) with the assistance of the homogenous Rh-TPPTS catalyst at atmospheric pressure and room temperature[22] as shown in Eq. (2):

$$C_5H_6O_4 + H_2 \xrightarrow{R.T, 1\,atm, Rh-TPPTS} C_5H_8O_4 \quad (2)$$

Considering that the hydrogenation reaction may not convert all the in situ generated $H_2$ to MSA, the "$H_2$-to-MSA conversion efficiency" ($\mu$), which ranges from 0% to 60%, is introduced for the different assessment scenarios. The amount of $H_2$ consumed ($\Phi_{H_2\_consumed}$) in the reaction is, therefore, a product of $\mu$ and the amount of $H_2$ produced on the cathode ($\Phi_{H_2\_produced}$):

$$\Phi_{H_2\_consumed} = \mu \times \Phi_{H_2\_produced} \quad (3)$$

The amount of $H_2$ collected after the reaction ($\Phi_{H_2\_collected}$) is:

$$\Phi_{H_2\_collected} = (1 - \mu) \times \Phi_{H_2\_produced} \quad (4)$$

The amount of MSA produced ($\Phi_{MSA\_produced}$) and IA consumed ($\Phi_{IA\_consumed}$) can therefore be determined:

$$\Phi_{MSA\_produced} = \frac{\Phi_{H_2\_consumed}}{M_{H_2}} \times M_{MSA} \quad (5)$$

$$\Phi_{IA\_consumed} = \frac{\Phi_{H_2\_consumed}}{M_{H_2}} \times M_{IA} \quad (6)$$

$M_{H_2}$, $M_{MSA}$ ($132.11\,g/mol$) and $M_{IA}$ ($130.18\,g/mol$) are the molecular masses of $H_2$, MSA, and IA, respectively.

### Scope definition

As shown in Fig. 1b, the scope of our assessment includes device manufacturing (photoelectrodes, catalysts, membrane, and encapsulation) and operation feedstocks (electrolyte and IA). Balance-of-system (BOS) components outside of the device boundary, e.g., piping, water containers, wiring, manifolds, etc., and the decommissioning process at the end of the service life are not considered in the current scope of the analysis.

The energetic cost of maintenance and human labor (i.e., work hours)[42] are also excluded from the scope of this study because the assessed technology is still at a single device scale; these two factors should be considered for the analysis of a larger-scale system.

The functional units defined in this study for the primary energy requirement of device components, the cumulative energy demand of device operation, and the energy obtained in the final products is MJ per m² of the PEC device area, which corresponds to the area of the photo absorber. To determine the levelized cost of energy of the products, the energy demand and the intrinsic energy content are calculated in the units of MJ/kg $H_2$ and MJ/kg MSA, respectively.

## Modeling approach

In this study, we follow the principles and framework of the Techno-Economic Assessment (TEA) & Life Cycle Assessment (LCA) Guidelines for $CO_2$ Utilization (Version 1.1) established by the Global $CO_2$ Initiative[43]. The guideline is a collection of existing ISO standards and guidelines that provides a specific protocol for multifunctional systems with several allocation methods for system inputs. Unlike the sub-division or system expansion allocation methods provided by ISO 14044[44] which lead to joint impact of the multifunctional system, a substitution method was applied in our study for product-specific impact assessments, and the amount of energy burden being avoided by coupled hydrogenation compared with conventional PEC device is considered. Life cycle inventory (LCI) values from the literature and the Ecoinvent database[45] were used to construct the target scenarios in Simapro v9.2.0. A detailed process flow diagram (PFD) with the primary level processing used in the Simapro calculation setup is shown in Fig. S4.

The energy return of the coupled PEC device was determined by performing a net energy analysis (NEA) to quantify the material and energy flows. NEA is a structured, comprehensive method to determine whether the energy yield of an energy production technology exceeds the energy required for exploitation (i.e., for extraction, processing, and transformation into a product, and delivery to the end-user)[46]. In NEA, the total amount of energy required and generated throughout the life cycle of a particular technology are compared to evaluate the energy implications and calculate the life cycle net energy balance. A technology with a negative net energy balance means that the energy required for its production is larger than that generated within its lifetime. The additional complexity and energy investment of renewable energy technologies vs. those of conventional energy generation methods can only be justified if a sufficiently favorable net energy balance is achieved.

A fundamental metric to quantify energy generation is the cumulative energy demand (CED). The CED of a product represents the direct and indirect energy used throughout the life cycle, including the energy consumed during the extraction of natural resources, preparation of upstream materials, manufacturing process, and disposal of the raw and auxiliary materials. For example, previous studies have presented CED values of 3535 MJ/m² for a SHJ PV cell[47] and 2110 MJ/m² for a silicon microwire PEC cell;[25] these values are used in our study. The CED of other components not previously reported in the literature is calculated based on their material and process requirements.

Due to the multifunctionality of our system, the allocation of energy flow between the two products (i.e., $H_2$ and MSA) was defined using the substitution method according to the guidelines[43]. In this method, one product is defined as the main product and the other as the sub-product. The levelized energy cost of the main product is determined after the energy earned from the sub-product is subtracted from the total energy investment.

When MSA is the target of the assessment, it is set to be the main product and $H_2$ is the sub-product. The energy allocated to MSA was calculated with Eq. (7).

$$CED_{kg\_MSA} = \frac{CED_{PEC} + CED_{H_2O} + E_{IA} - E_{H_2}}{\Phi_{MSA\_produced}} \quad (7)$$

$$E_{IA} = \Phi_{IA\_consumed} \times CED_{kg\_IA} \quad (8)$$

$CED_{kg\_MSA}$ is the CED of PEC-produced MSA in MJ/kg, $CED_{PEC}$ is the CED of the PEC device in MJ/m², $CED_{H_2O}$ is the CED of water consumption in MJ/m², $E_{IA}$ is the total energy consumption for IA in MJ/m², and $CED_{kg\_IA}$ is the CED of IA consumed in MJ/kg. The CED of PEC, $H_2O$, and IA were determined using specific regional data (e.g., electricity data from Germany's country mix) and described in the Life Cycle Inventory section. $E_{H_2}$ is the energy obtained from the collected $H_2$, calculated using the lower heating value (LHV) of 120 MJ/kg $H_2$:

$$E_{H_2} = \Phi_{H_2\_collected} \times LHV_{H_2} \quad (9)$$

Similarly, if $H_2$ is selected as the assessment target, the corresponding CED can be calculated accordingly. In this case, since the sub-product MSA is not a chemical fuel, the energy earned from MSA needs to be pre-determined using a benchmark method for industrialized MSA production instead of its lower heating value. In other words, the energy earned from co-producing MSA can be understood as the energy avoided to produce the same amount of MSA using the conventional method. Catalytic hydrogenation of IA is the current approach to produce MSA[13], which consumes 84 MJ/kg of MSA ($CED_{kg\_MSA,benchmark}$); the detailed energy demand of this process is discussed in the Life Cycle Inventory section below. The CED of the collected $H_2$ ($CED_{kg\_H_2}$, in MJ/kg) was therefore calculated using the following equations:

$$CED_{kg\_H_2} = \frac{CED_{PEC} + CED_{H_2O} + E_{IA} - E_{MSA}}{\Phi_{H_2\_collected}} \quad (10)$$

$$E_{MSA} = \Phi_{MSA\_produced} \times CED_{kg\_MSA,benchmark} \quad (11)$$

In the case that the PEC device is only producing $H_2$, the $H_2$-to-MSA conversion efficiency ($\mu$) is set to zero and $\Phi_{H_2\_collected} = \Phi_{H_2\_produced}$; Eq. (10) then simplifies to:

$$CED_{kg\_H_2} = \frac{CED_{PEC} + CED_{H_2O}}{\Phi_{H_2\_produced}} \quad (12)$$

## Life cycle inventory

The LCI of primary and cumulative energy requirements for device components, fabrication steps, and produced chemicals were obtained through different approaches. Device components and their fabrication steps are divided into four main components: synthesis of photoelectrodes and catalysts, fabrication of membrane, encapsulation, and other ancillary processes. The cumulative energy requirement was then calculated by adding the energy requirements for (i) raw materials production, (ii) fabrication processes, and (iii) operational feedstock. Data on several common materials (e.g., metals, plastics, common acids) and photovoltaic components (e.g., silicon wafer) that have already been previously investigated by life cycle analysis are available in the Ecoinvent database or the literature; they were therefore used directly in this study. For materials that are not available in the database or literature, we built the individual LCI for a specific synthesis process or used representative data on proxy materials (see SI Table S7). For fabrication processes that have not been evaluated before, e.g., the spray pyrolysis process of $BiVO_4$, we used thermodynamic modeling based on practical laboratory data and parameters to provide bounding estimates.

## Device components

The coupled PEC device investigated in this study adopts a silicon heterojunction (SHJ) solar cell without front metal contacts as the bottom absorber. Louwen et al. reported an LCA study on SHJ cells with an energy demand of 2110 MJ/m²[47]. We adapted this data for

single-sided metallization by removing half of the materials and energy consumed in the metallization process; only the back metal electrical contact is needed and the SHJ is directly interfaced with the $BiVO_4$ photoanode in our coupled device (see Fig. 1a). We also implement our region-specific data instead. The cumulative energy demand of the SHJ cell for our study was calculated to be 2530 MJ/m², and the detailed breakdown of the data is listed in SI Table S8. It should be noted that silver-based metallization has been identified to play a significant role in energy investment and the total cost of the SHJ cell due to the scarcity and high price of silver[48]. Some of the current manufacturing processes of SHJ cells have therefore been optimized using aluminum metallization instead, which brings a considerable material cost reduction[49], especially for large-scale implementations. This transition from silver to aluminum metallization has not been considered here.

In our coupled PEC device, the top absorber, $BiVO_4$-based photoanode, is assumed to be deposited directly on the ITO layer of the SHJ cell, which prevents the need for an additional glass substrate and reduces the complexity of the device. The material consumption and the fabrication steps considered were obtained from our previous report on spray-pyrolyzed $BiVO_4$[50]. The valve used to spray the $BiVO_4$ precursor solution has a power rating of 4 W[51], which corresponds to electricity consumption of 4.5 MJ/m². Continuous heating of the substrate (SHJ, in this case) is required to maintain a temperature of 450 °C during the deposition. After deposition, the SHJ/$BiVO_4$ samples are subjected to an additional heat treatment for 2 hours in a tube furnace at 450 °C in air. Unfortunately, there was no previous report available that documented the primary energy use of spray pyrolysis processes. One option is to directly use the parameters in our experiments, but these lab-scale values are not likely to be meaningful for large-scale production equipment. Instead, we used simple theoretical thermo-dynamic models to estimate the energy use for heating processes. Ideal insulation and control were assumed.

$$E_{h\_theoretical} = C_P \times M \times \triangle T \quad (13)$$

where $E_{h\_theoretical}$ is the amount of energy needed for heating, $C_p$ and $M$ are the specific heat and mass of the medium, respectively, and $\Delta T$ is the difference between the required temperature and the room temperature. $E_{h\_theoretical}$ represents the lower bounds of the actual primary energy requirements, i.e., assuming 100% thermal efficiency. The actual energy used for heating ($E_{h\_actual}$) is the theoretical value divided by the thermal efficiency ($\eta_{th}$). A thermal efficiency value of 50% was applied here, as also used in another study in the literature[25].

$$E_{h\_actual} = \frac{E_{h\_theoretical}}{\eta_{th}} \quad (14)$$

The energy used for the fabrication process (heating and pumping) is assumed to be supplied in the form of electricity. We estimate the primary energy demand for generating the required electricity by employing a primary energy-to-electricity conversion factor of 0.29[52]:

$$E_{h\_electricity} = \frac{E_{h\_actual}}{0.29} \quad (15)$$

During the spray pyrolysis, the pressure of the carrier gas $N_2$ drops from 3 bar to 1 bar through a 6 mm diameter tube. To estimate the $N_2$ consumption, we applied the Weymouth Equation[53,54] for high-pressure gas flow assuming steady-state adiabatic (isothermal) flow (see Supplementary Note 4). The flow rate of $N_2$ gas is $5.95 \times 10^{-4}$ m³/s which leads to the consumption of 595 L of $N_2$ for the whole duration of the spray pyrolysis process.

The oxygen and hydrogen evolution catalysts, i.e., cobalt phosphate (Co-$P_i$) and Pt, are electrodeposited onto $BiVO_4$ and solar-grade glass, respectively. The primary energy use of the electrodeposition

process for thin film coatings has been documented as 1.536 kWh/m²[55], which includes dipping (0.125 kWh/m²/minute), rinsing, drying, and calcination steps. We adapted this data to our electrodeposition process, which only involves a 15-minute dipping treatment. The estimated primary energy requirements for the electrodeposition of Co-$P_i$ and Pt are 20.1 MJ/m² and 13 MJ/m², respectively. A detailed overview of the amounts of materials and energy used for the photoelectrode fabrication and catalyst deposition is given in SI Table S9.

Finally, the energy demand for membranes, encapsulation, and other ancillary processes (i.e., miscellaneous chemicals, water pumping, environmental control, and cleaning of the manufacturing facilities) involved in the coupled PEC device is disaggregated from the net energy balance analysis of the PEC water splitting system reported by Zhai et al.[25]. The detailed LCI data are listed in SI Table S10.

### Itaconic acid (IA) feedstock

The IA feedstock in this study is considered to be mainly produced through a bio-fermentation method. Nieder-Heitmann et al. have reported an LCA study of plant-scale mass production of IA, which documented all the materials and emissions flow[56]. We took the same conditions for modeling in Simapro, and the cumulative energy demand ($CED_{kg\_IA}$) was determined as 8.42 MJ/kg IA. The concentration of IA in the electrolyte before being fed into the coupled PEC device was determined by assuming 60% $H_2$-to-MSA conversion efficiency to ensure a sufficient supply of reactant. The remaining unreacted IA, which depends on the $H_2$-to-MSA conversion efficiency, was assumed to be recycled and fed back into the device.

### Benchmark MSA production

No reports exist on the LCA and/or net energy assessment for MSA production. Therefore, a primary cumulative energy assessment was first performed in this study to determine the CED of MSA production through the currently deployed processes. MSA is conventionally synthesized from IA through conventional hydrogenation under controlled temperature and pressure[13]. Recently, the use of crude fermentation broth for electrochemical reduction of IA to MSA was also proposed to be an effective approach[39]. We considered both processes, and the process conditions and material consumption were obtained from the literature[13,39]. The detailed LCI data are listed in SI Table S11, and Simapro was used to compile the energy and material consumption under specific reaction conditions. Considering that this is the first time such an energy demand analysis of conventional MSA production is reported, an uncertainty analysis was also conducted (See Supplementary Note 5). The CED of MSA is 84 MJ/kg and 6800 MJ/kg for the hydrogenation and fermentation methods, respectively. Based on the much more favorable energy performance, we selected the conventional hydrogenation-produced MSA here as the benchmark product for further comparison.

### Water supply

The supply of deionized water (resistivity >1 MΩ cm at 25 °C) was assumed as the feedstock for our device. The energy consumed to deionize water was taken as 0.02 MJ/kg, as obtained from the Ecoinvent database. The flow rate was calculated based on 1 m² of cell area.

### Homogenous catalyst Rh-TPPTS

A rhodium TPPTS (Rh-TPPTS) homogenous catalyst with a concentration of 0.64 mM was considered for the hydrogenation reaction in the catholyte[57]. We assumed no operational loss or degradation of the catalyst. Considering that the thickness of the catholyte is 4 mm in our coupled PEC device, the amount of Rh-TPPTS used was calculated to be 0.01 g/m² of cell area.

### Performance metrics

Various metrics are used in this study to quantify the performance of the coupled PEC device. The normalized net energy balance (NNEB

in MJ/m²/year) is calculated as the annual energy input subtracted from the annual energy output:

$$NNEB = \frac{E_{H_2} + E_{MSA} - CED_{PEC} - E_{IA} - CED_{H_2O}}{t_{device}} \quad (16)$$

Another principal indicator of NEA is the energy return on energy investment (EROEI), which is defined as the ratio of the amount of energy produced in the form of useful energy carriers by a chain of processes exploiting a primary energy source to the total energy invested in finding, extracting, processing, and delivering that energy. In other words, EROEI is the energy output of the products divided by the energy input, taken over the entire device lifetime:

$$EROEI = \frac{E_{MSA} + E_{H_2}}{CED_{PEC} + E_{IA} + CED_{H_2O}} \quad (17)$$

Finally, the energy payback time (EPT) is the amount of time—typically in units of years—necessary for the device to operate so that it generates the same amount of energy as that consumed during its fabrication. It can be calculated by dividing the cumulative energy demand of the device by the net operational energy output per year:

$$EPT = \frac{CED_{PEC}}{\left(E_{MSA} + E_{H_2} - E_{IA} - CED_{H_2O}\right)/t_{device}} \quad (18)$$

## Data availability
All data supporting the findings of this study are available within the main text and the Supplementary Information file. Source data are provided with this paper.

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

## Acknowledgements

All authors acknowledge support from the Deutsche Forschungsgemeinschaft (DFG, German Research Foundation) under Germany's Excellence Strategy—EXC 2008/1 (UniSysCat)—390540038 and from the German Helmholtz Association—Excellence Network—ExNet-0024-Phase2-3. X.Z. acknowledges support from Zhandefu Materials (Zhenjiang) Ltd. The authors also thank Dr. Keisuke Obata for the experimental input from the coupled hydrogen production and hydrogenation as well as for valuable discussions.

## Author contributions

Conceptualization, X.Z. and F. F.A.; methodology, X.Z. and F.F.A.; investigation, X.Z.; visualization, X.Z. and F.F.A.; writing—original draft, X.Z.; writing—review & editing, M.S., R.S., R.v.d.K., and F.F.A.; supervision, R.S., R.v.d.K., and F.F.A.; funding acquisition, R.S., R.v.d.K., and F.F.A.

## Funding

## Competing interests
The authors declare no competing interests.
