## [Peer Review File · Nature Communications]

REVIEWER COMMENTS

Reviewer #1 (Remarks to the Author):

Review of “ Sustainable H₂ production and hydrogenation of chemicals in a coupled photoelectrochemical device – a life cycle net energy assessment”

The authors conducted a life cycle assessment of a photoelectrochemical (PEC) device designed to generate either hydrogen (H₂) or hydrogen and methyl succinic acid (MSA). The system boundary includes material and energy balances that occur during the operation of the device, and all associated upstream processes. Results include primary and cumulative energy demand, and are presented for a functional unit of one kilogram of hydrogen (at undefined temperature and pressure). The paper contributes novel insights into plausible manufacturing processes for a device based on a BiVO₄-based photoanode top absorber and a silicon heterojunction photovoltaic cell bottom absorber.

The authors provide new insights into the energy advantages of coupled H₂-MSA production over single product production from a PEC, and compare energy performance to competing technologies for H₂ production. The work is timely and original, and the methodology appears detailed and reproducible.

The paper's major limitation is the lack of rigor behind assumptions, and lack of robust sensitivity and uncertainty analysis, and therefore fails to meet the standard expected of life cycle assessments. The depth of discussion around results is also lacking. The authors have the opportunity to guide research and development on this exciting new technology, but fail to clearly describe and quantify technical targets. The benchmarking exercise is therefore limited in its utility. Minor and major revisions are noted in the following sections. The paper will be much stronger with the inclusion of an analysis of component and system level configurations and performance and the effects on results. In particular, the effects of catalyst degradation and need for replacement of components, a geographically agnostic analysis by way of sensitivity analysis (varying solar intensity and grid electricity), and breakdown of energy into primary energy source for the Germany case would be beneficial. The paper would also benefit from richer discussion around the size of facility that is practical for MSA production and the downstream processes that may impact life cycle energy demand.

Intro:

Pg. 3. Line 6. Please provide some sense of the mass production of MSA at a global or regional scale.

Pg 4. 84. Clarify what TPPTS is.

Methods:

Eq 2 on line 318 depicts a chemical reaction, but is a simplified expression neglecting the reaction conditions. This is misleading from a reactivity standpoint.

Using a conversion efficiency range of H₂ to MSA from 0 to 100% does not seem reasonable as 100% conversion has never been reported in literature or achieved in the author's reported experiments.

Not all readers will be familiar with the Guidelines for CO₂ Utilization from the Global CO₂ Initiative. It may be beneficial to note any differences with the standard ISO 14040 method for LCA. Tying this to the use of cumulative energy demand and net energy assessment definitions would reduce the need for extensive definitions as they currently are.

The authors use the energy embodied in MSA as a means of estimating the energy earned by the MSA product under a system expansion. (pg 20, line 398) It might be worth describing the logic behind the use of system expansion for allocating impacts, and potential pitfalls. For example, it implies perfect substitution of PEC MSA with existing MSA production. On this note, I might have missed it, but it is important to note that the energy intensity of conventional MSA production is also a finding of the study (benchmarking analysis), and therefore uncertain.

The EcoInvent database only has one value for deionized water. This and other limitations of the LCI should be discussed further, and perhaps captured in error bars. Other factors such as location (the electricity data), energy intensity of delivering water (is it deionized on site?), the primary energy consumption factor for electricity (pg 23, line 461), and solar intensity would be obvious choices for sensitivity analysis/error bars. In general, the lack of error bars is notable, given the novel configuration of the system that will have inherent uncertainty in it.

Results:

The resolution of schematic figure in 1a is poor and needs improving.

How might the start and stop of the PEC system with the sun affect performance? The PEC is obviously not running 24/7 and the 3.4 kWh/m²/day is an average over the day. At what point does the system no longer operate? Is a fan blowing H₂ to a collection point where it is compressed? How was the energy consumption of the compressors estimated?

Is coagulation of catalyst in water/suspension an issue in these systems? Has the architecture in 1a been demonstrated at any scale? How easily is H₂ and MSA separated? Some additional details on the design and research on design would be useful, as this seems to be a very important part of research and development for PEC.

Some sense of the quantity of water consumed per kg H₂ or by facility, and comparison with electrolyzer based H₂ production would be helpful. Readers may be surprised that the energy associated with transporting water to a facility making H₂ is minor.

The authors exclude the end of life phase of components, but they should have maintenance included as it affects the raw material and energy consumption over the system lifespan. The authors say replacement of materials is out of scope, but I do not think that is appropriate and the authors could demonstrate the effect even with approximations.

Pg 6 line 108 Rather than say “much less energy”, please provide a value or % .

The following statement needs further evidence: “For future optimization, the SHJ cell could be replaced with alternative absorbers that require less energy to produce.” Pg 6, lines 116-117. What is the value that should be targeted? Did the authors identify that performance target?

Fig 2 b, “cleaning” and “water pumping” are not well defined in the methodology.

It is likely that H₂ and MSA will need to be compressed and transported. At minimum, the authors should note at what pressure and temp H₂ is generated and whether that is different from H₂ from electrolyzers and SMR.

Further discussion around the impact of scale on results, or the methods inability to capture such impacts, is needed. How modular are these devices? What would be a logical scale based on facilities producing MSA? What is the water and land foot print of such a facility?

LCOE acronym may be confusing as I know it as the levelized cost of energy. For example, wind electrolyzer LCOE is reported as 9.1 MJ/kg on pg 9 line 152. The reference calls it total energy consumption.

Pg 12 lines 226-227. More details on why perovskite perovskite and polysulfone were selected would be helpful. The authors state that the process efficiency of these alternatives is outside the scope of analysis in the paragraph on pg 14 lines 254-260, but I think they could provide a review and rerun the analysis with more representative performance parameters.

In Figure 4, cases 3 and 4 drop off the chart after 5%, 5 years. Adding a summary statement about this result would explain the figure more clearly. Alternatively, referencing Figure 4 in the following paragraph starting on line 249 would explain the results clearly, instead of relying on the reader to make the connection independently.

There is a typo on lines 146 and 147 where STH efficiency and energy payback time were written backwards. The increase in payback time correlating with decreased efficiency makes sense, and so should then be written as 28.3, 17, and 8.5 years correlating with 3%, 5%, and 10% efficiency.

Hanna Breunig

Research Scientist, Lawrence Berkeley National Laboratory

Reviewer #2 (Remarks to the Author):

Review of: NCOMMS-22-33098-T

Sustainable H₂ production and hydrogenation of chemicals in a coupled photoelectrochemical device – a life cycle net energy assessment

This study presents a life cycle assessment of a theoretical photoelectrochemical device, in which water is split into oxygen and hydrogen, with the latter immediately reacting with itaconic acid (IA) over a homogeneous catalyst to produce methyl succinic acid (MSA). The study assesses the net energy benefit of such a device under different scenarios, including different fractions of in situ H₂ conversion to MSA, as well as different photoabsorbers (perovskite solar cells versus silicon heterojunction cells) and membrane separators (Nafion versus polysulfone).

The main finding is that with the use of typical components: (i) silicon heterojunction (SHJ) solar cells and (ii) Nafion membrane, the energy output from the generated H₂ cannot compensate the energy input required for the fabrication and operation of the device, unless very high solar-to-hydrogen conversion efficiencies and unrealistic device lifetimes are achieved. However, with the coupled hydrogenation reaction, the net energy balance shifts to much more favourable values. Hence, there is a clear case for coupling photoelectrochemical hydrogen generation with additional chemistries.

The life cycle analysis has been conducted thoroughly, with all assumptions/approximations clearly described and justified. It is a very interesting manuscript. The main criticisms are explained below; these must be addressed before publication.

Abstract

The abstract is written in a way that gives the impression that the results have been corroborated, whereas they are only predictive (in lines 280 – 282 the authors allude to being in the process carrying out experimental studies; however, the results of these are not presented and so can't really be taken into account). In the following two sentences of the abstract it should be made clear that the results are the output of a model:

“A negative net energy balance is [predicted to be] achieved when the device generates only hydrogen, but energy breakeven can already be achieved when a small ratio (~2%) of the generated hydrogen is used in situ for IA-19 to-MSA conversion. Moreover, the [simulated] coupled device produces MSA with much lower cumulative energy demand than conventional hydrogenation.”

Introduction

The introduction would benefit from an additional paragraph that summarizes the reports of experimental studies published to date on the hydrogenation of itaconic acid to methyl succinic acid,

since the chemistry has clearly already been published and is not the novel aspect of the work. For example, the hydrogenation of itaconic acid to methyl succinic acid over Rh-TPPTS catalyst is described in reference [39]. The novelty lies in the theoretical coupling of this chemistry with photoelectrochemical hydrogen production, the benefit of which is supported by the authors using life cycle analysis.

p. 3 - please define the abbreviation 'NEA'. A definition is provided on p. 18 but it would be better to have it where the abbreviation is first used.

Methods

p. 16 Please state unambiguously how the 'device area' is defined. Is it the area of the photoabsorber?

What is the exact purpose for metallization of the SHJ PV with silver or aluminium? Metallization was presumably applied to the back side of the PV (as can be inferred from Table S5)? Please expand on this, so that it is clearer.

Reviewer #3 (Remarks to the Author):

The manuscript by Zhang and co-workers describes a conceptual reaction scheme that employs a photoelectrochemical (PEC) device based on a BiVO₄/silicon heterojunction panel to perform H₂ evolution and the evolved H₂ is partially used in the hydrogenation of itaconic acid (IA) to methyl succinic acid (MSA) via a homogeneous catalyst. The authors calculated the net energy balance, which takes into account device fabrication and operation, and discovered that when H₂ and MSA co-evolve, the net energy balance can be positive, as opposed to when just H₂ is produced. A detailed calculation regarding the LCA of the PEC device is presented. However, I would like to have the following details on the IA hydrogenation process:

1) It is unclear why MSA production is chosen in this scheme. What is the size of the market for this chemical, and what is the predicted demand? How does it fare in comparison to the H₂ market? Given that it is intended that at least 2% of the H₂ produced be used to produce MSA to increase the potential of energy breakeven, what type of scenario or industry will this scheme be suitable for, and at what production scale?

2) Is the LCA analysis taking into account the device footprint/area preparation/installation?

3) What about the LCA of itaconic acid manufacturing, which is used as a feedstock in this process?

4) Is the energy demand for IA, MSA, and homogeneous catalyst separation and purification considered?

5) On line 281, the authors claimed that preliminary H₂-to-MSA conversion studies show conversion efficiency of up to 60%; may this preliminary data be included in the manuscript? This real data will strengthen the manuscript.

6) Why was the Rh-TPPTS catalyst selected for this scheme? Do the reaction conditions match well with the PEC device's working conditions? Could this hydrogenation process be done electrochemically by the electrocatalyst at the dark cathode?

7) Is this co-evolved H₂ and MSA process in a PEC device more favorable than the PV-electrolysis scheme?

Manuscript ID: NCOMMS-22-33098-T

Title: Sustainable H₂ production and hydrogenation of chemicals in a coupled photoelectrochemical device – a life cycle net energy assessment

REVIEWER REPORT(S): (Authors' response and associated changes in blue)

Reviewer #1 (Remarks to the Author):

Review of "Sustainable H₂ production and hydrogenation of chemicals in a coupled photoelectrochemical device – a life cycle net energy assessment"

The authors conducted a life cycle assessment of a photoelectrochemical (PEC) device designed to generate either hydrogen (H₂) or hydrogen and methyl succinic acid (MSA). The system boundary includes material and energy balances that occur during the operation of the device, and all associated upstream processes. Results include primary and cumulative energy demand and are presented for a functional unit of one kilogram of hydrogen (at undefined temperature and pressure). The paper contributes novel insights into plausible manufacturing processes for a device based on a BiVO₄-based photoanode top absorber and a silicon heterojunction photovoltaic cell bottom absorber.

The authors provide new insights into the energy advantages of coupled H₂-MSA production over single product production from a PEC and compare energy performance to competing technologies for H₂ production. The work is timely and original, and the methodology appears detailed and reproducible.

The paper's major limitation is the lack of rigor behind assumptions, and lack of robust sensitivity and uncertainty analysis, and therefore fails to meet the standard expected of life cycle assessments. The depth of discussion around results is also lacking. The authors have the opportunity to guide research and development on this exciting new technology but fail to clearly describe and quantify technical targets. The benchmarking exercise is therefore limited in its utility. Minor and major revisions are noted in the following sections. The paper will be much stronger with the inclusion of an analysis of component and system level configurations and performance and the effects on results. In particular, the effects of catalyst degradation and need for replacement of components, a geographically agnostic analysis by way of sensitivity analysis (varying solar intensity and grid electricity), and breakdown of energy into primary energy source for the Germany case would be beneficial. The paper would also benefit from richer discussion around the size of facility that is practical for MSA production and the downstream processes that may impact life cycle energy demand.

Response: First of all, we thank the reviewer for the critical evaluation of our work, and we are pleased to know that the reviewer considers our work as timely and original in delivering novel insights into the coupled H₂-to-MSA production with PEC device. We are also happy that the reviewer finds our methodology to be detailed and reproducible, and we greatly appreciate the valuable comments and suggestions. We address these comments in the specific point-by-point responses below, and we believe that the revised manuscript has been much improved with the addition of further analysis and discussion as suggested by the reviewer.

Introduction:

1.) Pg. 3. Line 66. Please provide some sense of the mass production of MSA at a global or regional scale.

Response: We appreciate the reviewer's comment regarding the need to provide the scale of MSA production. However, market information on MSA is not readily available since it is currently produced

at a limited amount for niche markets. Therefore, we tried to deduce the market size of MSA by considering the scale of itaconic acid, which is the main feedstock of MSA production, and the scale of succinic acid as a chemical product with a similar value chain as MSA. We have added the additional information and estimate to the Introduction section and as **Supplementary Note 1** in the supplementary sheet. Finally, we would like to point out that the hydrogenation of IA-to-MSA is chosen here only as a model reaction, and we expect that the coupled concept can be applied to other hydrogenation reactions with larger market size (e.g., acetone to isopropanol, CO₂ to hydrocarbons).

Associated changes to the manuscript:

- Page 3: "IA has been identified by the US Department of Energy as one of the twelve building blocks that possess the potential to be transformed subsequently to a number of high-value bio-based chemicals or materials.¹¹ MSA is a valuable chemical compound (with an estimated global market size of up to ~15,000 t – see Supplementary Note 1), whose derivatives are ubiquitously used as solvents in cosmetics,¹² polymer synthesis,¹³ binders in powder coatings,¹⁴ and in organic synthesis especially for pharmaceutical synthesis.^{15,16} Hydrogenation of IA to produce MSA has been reported using hydrogenation catalysts in a conventional hydrogenation reactor at 25-150 °C and 1-140 bar H₂.^{17-20"}

2.) Pg 4. 84. Clarify what TPPTS is.

Response: We thank the reviewer for bringing up this point. We have now added a clarified definition of TPPTS in the model description section.

Associated changes to the manuscript:

- Page 4: "A homogenous rhodium (Rh) trisodium 3,3',3''-phosphanetriyltri(benzene-1-sulfonate) (TPPTS) catalyst complex is dissolved in the catholyte for the hydrogenation of IA to MSA."

Methods:

3.) Eq 2 on line 318 depicts a chemical reaction but is a simplified expression neglecting the reaction conditions. This is misleading from a reactivity standpoint.

Response: We apologize that our initial expression of Eq. 2 neglected the reaction conditions. The reaction occurs at room temperature and atmospheric pressure; this information has now been added to Eq. 2 and in the main text.

Associated changes to the manuscript:

- Page 19 and 20: The *in situ* generated H₂ is then utilized to hydrogenate IA (C₅H₆O₄) to MSA (C₅H₈O₄) with the assistance of the homogenous Rh-TPPTS catalyst at atmospheric pressure and room temperature²² as shown in Eqn. (2):

4.) Using a conversion efficiency range of H₂ to MSA from 0 to 100% does not seem reasonable as 100% conversion has never been reported in the literature or achieved in the author's reported experiments.

Response: We agree that the range of H₂-to-MSA conversion efficiency should be limited to the range demonstrated in our laboratory experiments, which is up to 60%, although higher conversion—even 100%—is theoretically not impossible. We have made appropriate revisions to the manuscript text and

limited the data in **Table 2** to H₂-to-MSA conversion of $\leq 60\%$. Note that positive and negative ranges of errors in **Table 2** are now provided, and they correspond to the lower and higher cases in our uncertainty analysis, respectively, as further described in our response to **comment #7**.

Associated changes to the manuscript:

- Page 10 and 11: Only H₂-to-MSA conversion efficiency values of up to 60% are considered here; although higher conversion values are not theoretically impossible, this value (60%) represents the highest conversion efficiency already demonstrated in our preliminary experiments.
- Page 11 and 12: **Table 2**. Results of net energy balance assessment for co-producing H₂ and MSA under the base-case conditions of $\eta_{\text{STH}} = 5\%$, $t_{\text{device}} = 10$ years with H₂-to-MSA conversion efficiency from 0% to 60%. The error values provided for the normalized net energy balance and the cumulative energy demand (CED) of H₂ were obtained from the uncertainty analysis.

STH Efficiency	Longevity (years)	H ₂ -to-MSA conversion efficiency (μ)	Normalized Net Energy Balance (MJ/m ² /yr)	CED of H ₂ (MJ/kg)
5%	10	0%	-158±63	204±34
5%	10	20%	1,655±63	-982±42
5%	10	40%	3,469±63	-2,960±56
5%	10	60%	5,282±63	-6,915±84

- Page 20: Considering that the hydrogenation reaction may not convert all the *in situ* generated H₂ to MSA, the "H₂-to-MSA conversion efficiency" (μ), which ranges from 0% to 60%, is introduced for the different assessment scenarios.
- Page 27: The concentration of IA in the electrolyte before being fed into the coupled PEC device was determined by assuming 60% H₂-to-MSA conversion efficiency to ensure a sufficient supply of reactant.

5.) Not all readers will be familiar with the Guidelines for CO₂ Utilization from the Global CO₂ Initiative. It may be beneficial to note any differences with the standard ISO 14040 method for LCA. Tying this to the use of cumulative energy demand and net energy assessment definitions would reduce the need for extensive definitions as they currently are.

Response: We appreciate the suggestion from the reviewer. The Guidelines for CO₂ Utilization from the Global CO₂ Initiative is based on existing LCA ISO standards and guidelines as well as commonly applied assessment concepts and a collection of best practices. While ISO 14040 provides a general LCA guideline for any product systems, the Guidelines for CO₂ Utilization from the Global CO₂ Initiative is targeted at systems that feature additional chemical synthesis processes producing value-added products with the initial by-product as a feedstock. This is exactly what we consider in our study: (i) our system is a multifunctional one that generates two products, and (ii) the value-added product we generate (i.e., MSA) uses the initial product (i.e., H₂) as one of its main feedstocks. Therefore, the Guidelines for CO₂ Utilization from the Global CO₂ Initiative provides a suitable framework for our study.

A notable feature is regarding the approach used for input allocation. In ISO 14040, co-production is resolved by expanding or separating the system into several sub-systems. In the Guidelines for CO₂ Utilization from the Global CO₂ Initiative, a substitution method is introduced, which is adopted in our

study. With this method, the amount of burden being avoided by coupled hydrogenation compared with conventional PEC devices is considered. As a conceptual advantage, the substitution method conserves the causal interaction between processes by accounting for impacts in other life cycles.

We have now provided brief sentences in the revised manuscript that highlight the points mentioned above.

Associated changes to the manuscript:

- Page 21: "The guideline is a collection of existing ISO standards and guidelines that provides a specific protocol for multi-functional systems with several allocation methods for system inputs. Unlike the sub-division or system expansion allocation methods provided by ISO 14044⁴⁴ which lead to joint impact of the multifunctional system, a substitution method was applied in our study for product-specific impact assessments, and the amount of burden being avoided by coupled hydrogenation compared with conventional PEC device is considered."

6.) The authors use the energy embodied in MSA as a means of estimating the energy earned by the MSA product under a system expansion. (pg 20, line 398) It might be worth describing the logic behind the use of system expansion for allocating impacts, and potential pitfalls. For example, it implies perfect substitution of PEC MSA with existing MSA production. On this note, I might have missed it, but it is important to note that the energy intensity of conventional MSA production is also a finding of the study (benchmarking analysis), and therefore uncertain.

Response: Indeed, as the reviewer pointed out, we assume that PEC-generated MSA would be a perfect substitute for the existing MSA produced with a conventional method. We agree that the energy intensity of conventional MSA production is also a finding of this study that contains uncertainties. Therefore, an uncertainty analysis for the CED of MSA has been added to the supplementary information sheet.

Associated changes to the manuscript:

- Page 28: "Considering that this is the first-time such an energy demand analysis of conventional MSA production is reported, an uncertainty analysis was also conducted (See Supplementary Note 5)."
- Supplementary Note 5 has been added to the supplementary sheet, Pages 15 and 16:

"Supplementary Note 5. Uncertainty analysis for methyl succinic acid (MSA) production using a conventional method"

The input mass and energy data into MSA production is based on the information found in the literature, which was used to determine the cumulative energy demand (CED) of the conventional hydrogenation method of producing MSA. Considering the immaturity of manufacturing techniques and the fluctuation of operating conditions of MSA production, three main energy contributors were selected to address this inherent uncertainty: electricity consumption, hydrogenation catalysts, and itaconic acid (IA) feedstock. For electricity consumption of the production equipment, the thermal efficiency was assumed to be 30%, 50%, and 70%.¹⁸ The hydrogenation of IA to MSA has been reported with several hydrogenation catalysts, such as Raney Ni, Pd/C, Ru/C, etc.^{11,13,37} Iron-based catalysts are also expected to be developed for such a process, and this was considered for our analysis due to its low energy demand.²⁶ As for the IA feedstock, Nieder-Heitmann *et al* have reported a 30% uncertainty in their life cycle analysis.³⁸ Table S12 shows a summary of the input parameters and the percentage of uncertainty being introduced by each parameter to the individual and total MSA production process. The range of inputs uncertainty is shown in Figure S1 with error bars indicating the upper and lower limits of the energy demand."

Table S12. Life Cycle cumulative energy demand (CED) of inputs parameters in the unit of MJ/kg MSA production and output percentage of the uncertainty of MSA production

Input parameters	Cumulative energy demand (MJ/kg MSA) and uncertainty		
	Lower	Base case	Higher
Itaconic acid	5.83	8.33	10.83
Catalyst	4.60	31.10	161.87
Electricity	25.32	42.20	59.08
Total	64.80	84.18	234.32

Figure S1. The CED of MSA inputs parameters with error bars showing the range of uncertainties based on cases described in Table S12.

7.) The EcolInvent database only has one value for deionized water. This and other limitations of the LCI should be discussed further, and perhaps captured in error bars. Other factors such as location (the electricity data), energy intensity of delivering water (is it deionized on site?), the primary energy consumption factor for electricity (pg 23, line 461), and solar intensity would be obvious choices for sensitivity analysis/error bars. In general, the lack of error bars is notable, given the novel configuration of the system that will have inherent uncertainty in it.

Response: We fully agree with the reviewer's comment that the novel configuration of our proposed concept poses inherent uncertainties that need to be addressed in our study. We have now included an uncertainty analysis in the revised manuscript. For the device-level study, **Table S2** has been added to show the uncertainty estimates based on three different scenarios involving material choices and energy use in the fabrication process. The accompanying discussion has also been added as **Supplementary Note 2**. Error bars have been added accordingly to the data shown in **Figure 2(b)** and **Table 2**, based on the different scenarios.

For uncertainty factors beyond the device-level study, such as water transportation, gas handling, product separation, and solar intensity, a preliminary system-level analysis for a larger scale (100 m²) coupled PEC system has been added as **Supplementary Note 3**, and the corresponding uncertainty considerations are listed in **Table S3**. As a regional relevant uncertainty, different solar intensities were used for lower, base, and higher cases. The de-ionized water used for the electrolytes is indeed

considered to be treated on-site using grid water through reverse osmosis (RO) treatment, and the relevant energy demand for water delivery was estimated based on the water consumption and the distance for delivery. A brief discussion of the gas handling system has also been added with an estimation of the energy use.

Associated changes to the manuscript:

- Page 8: error bars were added to Figure 2(b), presenting the uncertainty of device material and fabrication LCI data:

Figure 2 | Cumulative energy demand of the coupled PEC device. (a) Energy distribution pie-chart of our coupled PEC device indicating the contribution from the various components. Photoelectrodes material and fabrication consume the most of energy (70.4%) and SHJ bottom absorber is the most energy-intensive component. **Errors bars for cell fabrication and materials correspond to uncertainty analysis.**

- Page 11 and 12: **Table 2.** Results of net energy balance assessment for co-producing H₂ and MSA under the base-case conditions of $\eta_{STH} = 5\%$, $t_{device} = 10$ years **with H₂-to-MSA conversion efficiency from 0% to 60%. Positive and negative ranges of errors corresponding to lower and higher cases from uncertainty analysis.**

STH Efficiency	Longevity (years)	H ₂ -to-MSA conversion efficiency (μ)	Normalized Net Energy Balance (MJ/m ² /yr)	CED of H ₂ (MJ/kg)
5%	10	0%	-158±63	204±34
5%	10	20%	1,655±63	-982±42
5%	10	40%	3,469±63	-2,960±56
5%	10	60%	5,282±63	-6,915±84

- Supplementary Note 2 has been added to the supplementary sheet, Page 5:

Supplementary Note 2. Uncertainty analysis for the device-level study considering three different scenarios (lower, base, and higher cases)

"Since the coupled PEC concept is at a very early stage of research at the device level, the life cycle inventory data used during its fabrication and operation has unavoidable uncertainties. This uncertainty analysis addresses this limitation by including variations in material choices and fabrication parameters under three assumed cases as shown in Table S2. The same top and bottom absorbers were used in all cases since this corresponds to our research target and the specific preliminary demonstration device in our lab. Other material choices were made based on their respective primary energy demands and categorized into lower and higher cases. For the cumulative energy demand of the fabrication process, uncertainty was introduced to the energy usage based on different assumptions of thermal and electricity conversion efficiency."

- Page 5: "Table S2. Assumptions for lower, base, and higher cases of uncertainty analysis at the device level."

Category	Component	Lower case	Base case	Higher case
Material choices	Bottom absorber	SHJ cell	SHJ cell	SHJ cell
	Top absorber	BiVO ₄	BiVO ₄	BiVO ₄
	Photocathode catalyst	Cobalt	Platinum	Platinum
	Photoanode catalyst	No catalyst	Co-Pi	Iridium (Ir)
	Chamber	3 mm PVC	5 mm PVC	5 mm Polycarbonate
	Membrane	30 μ m Nafion	50 μ m Nafion	70 μ m Nafion
	Homogenous catalyst	Nickel	Rh-TPPTS	Pd/C
Fabrication	Thermal efficiency	30%	50%	70%
	Electricity conversion efficiency	0.27	0.29	0.33

- Page 14: "In order to evaluate the impact of parameters beyond the device level (e.g., balance-of-system, regional uncertainties, utility delivery, product handling and separation), a preliminary system-level analysis was conducted considering a 100 m² (land area) coupled PEC system. The daily operation time of the system is assumed to be 12 hours, and the production of hydrogen gas and MSA occurs immediately upon exposure to sunlight.^{6,30,31} Detailed discussion of additional parameters can be found in Supplementary Note 3, and the breakdown of the cumulative energy demand for the overall system is shown in Table S4. As expected, the inclusion of additional system-level inputs increases the required H₂-to-MSA conversion efficiency for energy breakeven. Under the base case scenario, a 100 m² scale coupled PEC system requires 7.5% H₂-to-MSA conversion for energy breakeven. Although this is almost four-fold higher than the requirement from the device-level analysis, the 7.5% required H₂-to-MSA

conversion is still well within the feasibility range considering that H₂-to-MSA conversion of up to 60% has been demonstrated in our preliminary experiments.”

- Supplementary Note 3 has been added to the supplementary sheet, Pages 6, 7, and 8:

“Supplementary Note 3. Preliminary system-level analysis for a 100 m² coupled PEC system

To investigate the additional energy demand from factors beyond the PEC device itself (e.g., balance-of-system), a preliminary system-level analysis was also conducted. We considered and adapted the technical and engineering designs of a recently reported 100 m² photochemical water splitting system in Japan.²⁷ A single PEC device has a dimension of 295 mm × 250 mm and 625 cm² photoactive area. One panel consists of 48 devices, and the overall 100 m² system consists of 33¹/₃ panels of 3 m² each. The total light absorbing area is 70 m², while the remaining 30 m² of the land area is attributed to row spacing.

De-ionized water is used for the electrolytes (as also the case in our laboratory experiment), and a large amount of water feedstock is assumed to be treated on-site using grid water with reverse osmosis (RO) process.²⁸ The energy required for water pumping from the distribution station to the prospective system is calculated based on the average water transportation consumption. The energy demand depends on the distance and usage of water while the specific energy use for water pumping is based on the report of Plappally and Lienhard.²⁹ The average water delivery distance is assumed to be 10 km, and the daily water consumption is calculated based on the day on which peak generation occurs within the year, i.e., summer solstice (7.17 kWh/m²/day). Gas handling consists of a blower, dryer, and compressor which pressurizes H₂ to 300 psi at a temperature of 20 °C for delivery to end-users through pipelines. Sathre *et al* reported in their LCA of large-scale PEC system that the energy demand for gas handling accounts for 6.7% of the total primary energy investment and 34.7% of annual operational energy demand.³⁰ The same considerations are used in our study.

After the coupled PEC hydrogenation reaction takes place in the PEC device at a particular H₂-to-MSA conversion efficiency, the catholyte contains a mixture of IA, Rh-TPPTS, H₂ gas, and MSA, which is delivered to a separating unit. Compressed H₂ gas is collected and delivered to pipelines, and the remaining liquid solution is sent to another separating unit for MSA extraction. Recent separation techniques, such as micellar-enhanced ultrafiltration and cloud point extraction,^{31,32} have been reported for the separation of IA, MSA, and the homogenous catalyst. Conventionally, the catalyst is filtered off and suitable acids (hydrochloric acid, sulfuric acid) is added to extract MSA from its metal salt.⁹ However, these processes are not yet commercialized, and for our system-level analysis, we consider the separation process of succinic acid as a proxy process for MSA separation.³³ As reported the downstream separation and purification contribute to 16% of the total cost in the SA production line. The same ratio was taken for the energy demand of MSA separation in our study to estimate the CED of this process.

The replacement interval of the device components is assumed to be once per year and 10% of device components being replaced.³⁰ At the end of the system service time, 10% capital energy is required for decommission.³⁴

Uncertainties were introduced to the fixed and O&M components of the system, according to cases listed in Table S3. As a regional relevant uncertainty, different solar intensities across Germany (Hamburg, German average, and Munich) were used for the lower, base, and higher cases.

- Supplementary information sheet Page 7:

Table S3. Assumptions for lower, base, and higher cases of uncertainty analysis at large level.

System components	Energy demand (MJ/m ²)		
	Lower case	Base case	Higher case
Fixed energy demand			
PEC device	3,434	3,834	4,462
Gas handling	160.8	257.3	522.4
Decommission	179.7	409.1	747.7
Separation unit	359.5	651.7	996.9
O&M energy demand			
Replacement	3,434	3,834	4,462
IA usage	3,425.7	4,076.8	4,361.4
Water usage	3.1	3.7	4.0
Water delivering	0.03	0.07	0.13
Gas handling	0.9	1.3	1.5
MSA separation	686.4	1,260.9	1,765.8

- Supplementary information sheet Page 8:

Table S4. Cumulative energy demand (CED) of inputs parameters for preliminary scaled-up analysis for a 100 m² coupled PEC system.

System components	Unit	Lower case	Base case	Higher case
Fixed energy demand				
PEC device	MJ/m ²	3,434	3,834	4,462
Gas handling	% of capital demand	4.7%	6.7%	11.7%
Decommission	% of capital demand	5%	10%	15%
Separation unit	% of capital demand	10%	16%	20%
O&M energy demand				
Annual replacement	% of PEC device	5%	10%	15%
CED of IA feedstock	MJ/kg	5.9	8.4	10.9
Water usage	kg/day/m ²	0.16	0.18	0.20
Water delivering	kWh/m ³ ·km	0.002	0.005	0.007
Gas handling	% of H ₂ production	29%	35%	36%
MSA separation	% of coupled process	10%	16%	20%
Solar intensity	kWh/m ² /day	2.9	3.5	3.7

Results:

8.) The resolution of a schematic figure in 1a is poor and needs improving.

Response: We thank the reviewer for bringing this up. The resolution of **Figure 1a** has now been improved as shown below.

Associated changes to the manuscript:

- Page 5: Figure 1a has been updated as shown below.

9.) How might the start and stop of the PEC system with the sun affect performance? The PEC is obviously not running 24/7 and the 3.4 kWh/m²/day is an average over the day. At what point does the system no longer operate? Is a fan blowing H₂ to a collection point where it is compressed? How was the energy consumption of the compressors estimated?

Response: Indeed, our coupled PEC system will not operate 24/7 as it is subjected to sunlight availability. The relationship between the operating performance of our PEC system and any start-up and shutdown effect is expected to be similar to a PEC system that generates only H₂, and we refer to studies reported in the literature (Sathre, R. et al. *Energy & Env. Sci.* 7, 2014, 3264; Kistler, T. A., Um, et al. *Journal of The Elec. Society* 167, 2020, 66502; Nishiyama, H. et al. *Nature* 598, 2021, 304). In our case, we assume 12 hour-operation and the device will start generating H₂ (and MSA) upon exposure to sunlight. A brief explanation of the large-scale system operation has been added. As for the H₂ delivery and compression, we have now considered these processes in the system-level analysis as discussed in our response to **comment #7**.

Associated changes to the manuscript:

- Page 14: "The daily operation time of the system is assumed to be 12 hours and production of hydrogen gas (and MSA) occurs immediately upon exposure to sunlight."^{6,30,31}
- Please refer to our response to **comment #7**.

10.) Is coagulation of catalyst in water/suspension an issue in these systems? Has the architecture in 1a been demonstrated at any scale? How easily is H₂ and MSA separated? Some additional details on the design and research on design would be useful, as this seems to be a very important part of research and development for PEC.

Response: As the Rh-TPPTS catalyst as well as IA and MSA are highly soluble in aqueous solution, coagulation is not expected to be an issue. The architecture in Figure 1a has indeed been demonstrated in our laboratory proof-of-concept experiments; the results are currently being considered for publication in another journal (see our response to **comment #5 of Reviewer 3**). However, in our experiments, we only performed product quantification without separating the products and the catalyst. The separation process has been reported in the literature using micellar-enhanced ultrafiltration and cloud point extraction (Schmidt et al. *Ind. Eng. Chem. Res.* 58, 2019, 2445; Schwarze et al. *Chem. Ing. Tech.* 93, 2021, 31). However, since these processes are not yet industrialized, we consider the separation

process of succinic acid (Efe et al. Biomass and Bioenergy, 56, 2013, 479) as a proxy in our system-level analysis (see our response to **comment #7**). We have now added this information to the system-level analysis in **Supplementary Note 3**.

Associated changes to the manuscript:

- Please refer to our response to comment #7.

11.) Some sense of the quantity of water consumed per kg H₂ or by facility, and comparison with electrolyzer-based H₂ production would be helpful. Readers may be surprised that the energy associated with transporting water to a facility making H₂ is minor.

Response: We thank the reviewer for the suggestion. The use and transport of water to a facility were not initially considered as only device-level analysis was performed. This has now been considered in the scale-up analysis as discussed in our response to **comment #7**. Indeed, as the reviewer pointed out, the energy associated with the use and transport of water to the facility is minor (< 0.1% of the total fixed and O&M energy needs).

Associated changes to the manuscript:

- Please refer to our response to comment #7.

12.) The authors exclude the end-of-life phase of components, but they should have maintenance included as it affects the raw material and energy consumption over the system lifespan. The authors say replacement of materials is out of scope, but I do not think that is appropriate and the authors could demonstrate the effect even with approximations.

Response: As mentioned previously, we indeed did not consider the maintenance or replacement of materials. We agree with the reviewer that this is important. We have now included the end-of-life and maintenance in the system-level analysis to make the life cycle more comprehensive (SI sheet Supplementary Note 3). The maintenance of the device components is assumed to be originated from the annual replacement of 10% of the device components. This assumption has also been used in a life cycle analysis of the PEC water splitting system (Sathre, R. et al. Energy & Env. Sci. 7, 2014, 3264). The end-of-life treatment for the device is added based on the assumption that 10% capital energy is required for decommissioning, as suggested by the National Energy Technology Laboratory (James, R. NETL, US DOE, 2011; Robert, E. NETL, 2010). System performance metrics have been recalculated with additional inputs and the resulting changes have now been discussed in the results and conclusion section. **Figure 1(b)**, which illustrates the system boundary, has been updated by including maintenance and end-of-life treatment blocks into consideration, and several corresponding sentences have been added to the caption.

Associated changes to the manuscript:

- Page 4 and 5: "A periodic maintenance and end-of-life treatment are included in the preliminary system-level assessment. For maintenance, it is assumed that 10% of the device components are replaced annually⁶. As for the end-of-life treatment, National Energy Technology Laboratory (NETL) published a summary of the method used in their series of energy system life cycle assessments which assume that decommissioning requires 10% of the capital energy used for the initial construction of the system.^{23"}
- Page 6: Figure 1(b) has been modified as shown below.

Figure 1 | Considerations used in the life cycle net energy assessment in this study. (a) Schematic drawing of the coupled photoelectrochemical device considered in this study, in which hydrogen production and hydrogenation of itaconic acid (IA) to methyl succinic acid (MSA) occur in the catholyte chamber. (b) Simplified process flow diagram of the coupled PEC device for hydrogen production and hydrogenation of IA to MSA, starting from raw material extraction to the end-of-life treatment. The dashed line indicates the cradle-to-gate system boundary.

- Supplementary information sheet Page 7: "The replacement interval of the device components is assumed to be once per year, and 10% of device components is replaced.³⁰ At the end of the system service time, 10% capital energy is required for decommission.³⁴"

13.) Pg 6 line 108 Rather than say "much less energy", please provide a value or %.

Response: We appreciate the suggestion from the reviewer, and a percentage value has now been provided to compare the energy consumption for material and fabrication.

Associated changes to the manuscript

- Page 6: The material required for the SHJ cell is the main contributor, while the fabrication process consumes 71% lower energy, as shown in Figure 2b.

14.) The following statement needs further evidence: "For future optimization, the SHJ cell could be replaced with alternative absorbers that require less energy to produce." Pg 6, lines 116-117. What is the value that should be targeted? Did the authors identify that performance target?

Response: We did not include any target value since the overall energy analysis for the coupled production device is actually already quite positive. However, we have now included comparison values for other emerging absorbers that might be considered as alternatives: silicon microwire and perovskite absorbers.

Associated changes to the manuscript:

- Page 7: "For instance, the cumulative energy demand of silicon microwire²⁵ and perovskite absorbers²⁶ have been reported to be more competitive at 661 MJ/m² and 779 MJ/m², respectively. Note, however, that they are not yet commercialized, and the energy demand is based on laboratory scale data."

15.) Fig 2 b, “cleaning” and “water pumping” are not well defined in the methodology.

Response: We apologize for missing to defining the terms “cleaning” and “water pumping” in the main text. The definition has now been added to the methodology section.

Associated changes to the manuscript:

- Page 27: “Finally, the energy demand for membranes, encapsulation, and other ancillary processes (i.e., miscellaneous chemicals, water pumping, environmental control and cleaning of the manufacturing facilities) involved in the coupled PEC device is disaggregated from the net energy balance analysis of the PEC water splitting system reported by Zhai et al.²⁵”

16.) It is likely that H₂ and MSA will need to be compressed and transported. At minimum, the authors should note at what pressure and temp H₂ is generated and whether that is different from H₂ from electrolyzers and SMR.

Response: The compression and transportation of products were not considered in the device-level analysis. We have now added this consideration to our system-level analysis, as described in our response to **comment #7**.

Associated changes to the manuscript:

- Please refer to our response to comment #7.

17.) Further discussion around the impact of scale on results, or the methods inability to capture such impacts, is needed. How modular are these devices? What would be a logical scale based on facilities producing MSA? What is the water and land footprint of such a facility?

Response: We thank the reviewer’s comment on the issue of the PEC scale as this is very important in our efforts of establishing practical deployment of the device. The technical and engineering parameters at a larger scale (including the water and land footprint) have been addressed in our preliminary system-level analysis, as discussed in our response to **comment #7**.

Associated changes to the manuscript:

- Please refer to our response to comment #7.

18.) LCOE acronym may be confusing as I know it as the levelized cost of energy. For example, wind electrolyzer LCOE is reported as 9.1 MJ/kg on pg 9 line 152. The reference calls it total energy consumption.

Response: We thank the reviewer for pointing out this misuse of the LCOE term. All instances in the manuscript where the term was used have now been corrected with cumulative energy demand (CED).

Associated changes to the manuscript:

- Page 10: “H₂ produced by coupling wind electricity and electrolyzers is the most energy-efficient method with the CED of H₂ at only 9.1 MJ/kg.⁷ The most common approach used in the industry is steam methane reforming (SMR) with the CED of H₂ at 183 MJ/kg H₂.²⁷”
- Page 10: “The use of grid electricity (which is composed of a mix of different sources) and electrolyzers has been reported to yield a CED of H₂ in the same range as that using SMR.²⁹ The

CED of H₂ generated from PEC water splitting varies, based on the large variation of STH efficiency and device longevity.”

19.) Pg 12 lines 226-227. More details on why perovskite perovskite and polysulfone were selected would be helpful. The authors state that the process efficiency of these alternatives is outside the scope of analysis in the paragraph on pg 14 lines 254-260, but I think they could provide a review and rerun the analysis with more representative performance parameters.

Response: We appreciate the reviewer’s comment regarding the optimization analysis. Our choices of alternatives have now been supported with added performance parameters of the perovskite-perovskite absorber and the polysulfone membrane. **Figure 4** has also been updated after recalculating the results with added energy demand for replacement at a 5-year interval for the perovskite-perovskite absorber due to its limited lifetime as compared with SHJ.

Associated changes to the manuscript:

- Page 15: “Recently, Zhao *et al.* reported that the lifetime of perovskite absorbers is predicted to reach 5 years with a 17.4% efficiency.³⁴ Although a higher energy demand is expected due to more frequent replacement, perovskites still represent the most promising absorber alternative, especially since intense ongoing research efforts are likely to lead to further improvements in the material’s stability. As a suitable membrane material, PSF has been proposed as a concrete alternative for Nafion with comparable performance.³⁵ According to the life cycle analysis from Yadav *et al.*, the impact of PSF on the environment is lower than that of other membrane materials.³⁶”
- Page 15: “Note that an additional 778.6 MJ/m² energy demand (CED of perovskite absorber) is included when the investigated longevity is > 5 years since the perovskite lifetime is assumed to be 5 years.”
- Page 15 and 16: “Figure 4 shows the required H₂-to-MSA conversion efficiencies to achieve energy breakeven for the four different cases of devices under various STH efficiencies and device longevity conditions. When the longevity is 5 years, the substitution of the bottom absorber from SHJ to perovskite reduces the required conversion efficiency by more than 50% (case 1 vs. 3) while the required conversion efficiency remains more or less unchanged when the membrane is changed from Nafion to polysulfone (case 1 vs. 2). This is not surprising since the reduction of the primary energy demand is much higher when replacing the bottom absorber instead of the membrane (see Table S6). However, when the longevity is 10 years, the replacement of perovskite-perovskite absorber increases the energy demand. As a result, the required conversion efficiency drops by 39.2% which is less than that in 5-year scenarios.”
- Page 16 and 17: Figure 4 has been updated as shown below.

Figure 4 | Energetic impact of substituting components of the coupled PEC device. Threshold H₂-to-MSA conversion efficiency for energy to breakeven in four cases of using different bottom absorber and membrane combinations in our coupled PEC device. No conversion efficiency is shown for case 3 (Perovskite_Nafion) and case 4 (Perovskite_Polysulfone) when the STH is 10% and longevity is 5 years since H₂ production itself is sufficient to pay back the energy investment, i.e., no H₂-to-MSA conversion is needed.

20.) In Figure 4, cases 3 and 4 drop off the chart after 5%, 5 years. Adding a summary statement about this result would explain the figure more clearly. Alternatively, referencing Figure 4 in the following paragraph starting on line 249 would explain the results clearly, instead of relying on the reader to make the connection independently.

Response: We thank the reviewer for the suggestion. A summary statement about cases 3 and 4 is added to the figure caption to make our content more reader-friendly. Since **Figure 4** has also been updated to address **comment #19**, please refer to our response and the associated changes listed above.

21.) There is a typo on lines 146 and 147 where STH efficiency and energy payback time were written backwards. The increase in payback time correlating with decreased efficiency makes sense, and so should then be written as 28.3, 17, and 8.5 years correlating with 3%, 5%, and 10% efficiency.

Response: We thank the reviewer for pointing out this typo. The sentence has been corrected with corresponding values in the right order.

Associated changes to the manuscript:

- Page 9: "The energy payback time obviously decreases with increasing η_{STH} : 28.3, 17, and 8.5 years were needed for devices with η_{STH} of 3%, 5%, and 10%, respectively."

Hanna Breunig
Research Scientist, Lawrence Berkeley National Laboratory

Reviewer #2 (Remarks to the Author):

Review of: NCOMMS-22-33098-T

Sustainable H₂ production and hydrogenation of chemicals in a coupled photoelectrochemical device – a life cycle net energy assessment

This study presents a life cycle assessment of a theoretical photoelectrochemical device, in which water is split into oxygen and hydrogen, with the latter immediately reacting with itaconic acid (IA) over a homogeneous catalyst to produce methyl succinic acid (MSA). The study assesses the net energy benefit of such a device under different scenarios, including different fractions of in situ H₂ conversion to MSA, as well as different photoabsorbers (perovskite solar cells versus silicon heterojunction cells) and membrane separators (Nafion versus polysulfone).

The main finding is that with the use of typical components: (i) silicon heterojunction (SHJ) solar cells and (ii) Nafion membrane, the energy output from the generated H₂ cannot compensate the energy input required for the fabrication and operation of the device, unless very high solar-to-hydrogen conversion efficiencies and unrealistic device lifetimes are achieved. However, with the coupled hydrogenation reaction, the net energy balance shifts to much more favourable values. Hence, there is a clear case for coupling photoelectrochemical hydrogen generation with additional chemistries. The life cycle analysis has been conducted thoroughly, with all assumptions/approximations clearly described and justified. It is a very interesting manuscript. The main criticisms are explained below; these must be addressed before publication.

Response: We thank the reviewer for the critical evaluation of our work, and we are grateful that the reviewer finds our study to be thorough and very interesting. We appreciate the valuable criticisms from the reviewer, which we address point-by-point below. At this point, we also would like to mention that our study is based on experimental results from our laboratory device (i.e., not fully theoretical), but we indeed extended the results to a theoretical service lifetime for life-cycle analysis.

Abstract

1.) The abstract is written in a way that gives the impression that the results have been corroborated, whereas they are only predictive (in lines 280 – 282 the authors allude to being in the process carrying out experimental studies; however, the results of these are not presented and so can't really be taken into account). In the following two sentences of the abstract it should be made clear that the results are the output of a model:

"A negative net energy balance is [predicted to be] achieved when the device generates only hydrogen, but energy breakeven can already be achieved when a small ratio (~2%) of the generated hydrogen is used in situ for IA-19 to-MSA conversion. Moreover, the [simulated] coupled device produces MSA with much lower cumulative energy demand than conventional hydrogenation."

Response: We thank the reviewer for raising up this important point. Our abstract has been updated as suggested.

Associated changes to the manuscript:

- Page 1: "A negative net energy balance is predicted to be achieved when the device generates only hydrogen, but energy breakeven can already be achieved when a small ratio (~2%) of the generated hydrogen is used in situ for IA-to-MSA conversion. Moreover, the simulated coupled device produces MSA with much lower cumulative energy demand than conventional hydrogenation."

Introduction

2.) The introduction would benefit from an additional paragraph that summarizes the reports of experimental studies published to date on the hydrogenation of itaconic acid to methyl succinic acid, since the chemistry has clearly already been published and is not the novel aspect of the work. For example, the hydrogenation of itaconic acid to methyl succinic acid over Rh-TPPTS catalyst is described in reference [39]. The novelty lies in the theoretical coupling of this chemistry with photoelectrochemical hydrogen production, the benefit of which is supported by the authors using life cycle analysis.

Response: We appreciate the suggestion from the reviewer. We have now added a brief reference to current reports on the hydrogenation of itaconic acid to methyl succinic acid.

Associated changes to the manuscript:

- Page 3: "Hydrogenation of IA to produce MSA has been reported using hydrogenation catalysts in a conventional hydrogenation reactor at 25-150 °C and 1-140 bar H₂.¹⁷⁻²⁰"

3.) p. 3 - please define the abbreviation 'NEA'. A definition is provided on p. 18 but it would be better to have it where the abbreviation is first used.

Response: We agree with the reviewer that a definition for the abbreviation needs to be provided where it is first used in the manuscript. Indeed, we have provided the definition of 'NEA' on page 2 line 45 where it is first mentioned:

*"Despite the impressive progress that has been made in this field, several techno-economic analyses (TEA) and **net energy assessments (NEA)** have indicated that the PEC approach is still not energetically and economically competitive for large-scale implementation."*

Methods

4.) p. 16 Please state unambiguously how the 'device area' is defined. Is it the area of the photo absorber?

Response: We thank the reviewer for the suggestion. Indeed, the device area refers to the photo absorber area in our study. We have clarified this in the revised manuscript.

Associated changes to the manuscript:

- Page 21: "The functional units defined in this study for the primary energy requirement of device components, the cumulative energy demand of device operation, and the energy obtained in the final products is MJ per m² of the PEC device area, which corresponds to the area of the photo absorber."

5.) What is the exact purpose for metallization of the SHJ PV with silver or aluminium? Metallization was presumably applied to the back side of the PV (as can be inferred from Table S5)? Please expand on this, so that it is clearer.

Response: We apologize for not clearly describing this in our initial manuscript. Metallization was performed in solar cells to provide electrical contacts. In commercial SHJ PV cell, as two terminals are needed for the positive and negative contacts, metallization is performed on both the front and back sides of the cell. However, in our device, the front metal contact is not needed as the SHJ is directly interfaced with the BiVO₄ photoanode (see **Figure 1a**). Therefore, as shown in **Table S8**, we only consider

a single-side metallization on the back side of the SHJ absorber, and the energy demand would be reduced by half. We have added a clarification sentence to the manuscript.

Associated changes to the manuscript:

- Page 25: "We adapted this data for single-sided metallization by removing half of the materials and energy consumed in the metallization process; only the back metal electrical contact is needed and the SHJ is directly interfaced with the BiVO₄ photoanode in our coupled device (see Figure 1a). We also implement our region-specific data instead."

Reviewer #3 (Remarks to the Author):

The manuscript by Zhang and co-workers describes a conceptual reaction scheme that employs a photoelectrochemical (PEC) device based on a BiVO₄/silicon heterojunction panel to perform H₂ evolution and the evolved H₂ is partially used in the hydrogenation of itaconic acid (IA) to methyl succinic acid (MSA) via a homogeneous catalyst. The authors calculated the net energy balance, which takes into account device fabrication and operation, and discovered that when H₂ and MSA co-evolve, the net energy balance can be positive, as opposed to when just H₂ is produced. A detailed calculation regarding the LCA of the PEC device is presented. However, I would like to have the following details on the IA hydrogenation process:

1) It is unclear why MSA production is chosen in this scheme. What is the size of the market for this chemical, and what is the predicted demand? How does it fare in comparison to the H₂ market? Given that it is intended that at least 2% of the H₂ produced be used to produce MSA to increase the potential of energy breakeven, what type of scenario or industry will this scheme be suitable for, and at what production scale?

Response: We thank the reviewer for his/her critical evaluation of our work. We have added more background discussion of MSA in the introduction section and as a supplementary note. We refer the reviewer to our response to **comment #1 of Reviewer 1**. In addition, we conducted a preliminary system-level analysis that addresses a larger scale, which is discussed in our response to **comment #7 of Reviewer 1**. We also would like to point out again that the hydrogenation of IA-to-MSA is chosen only as a model reaction for our coupled PEC hydrogenation concept. Although it is beneficial from the energetic point of view as shown in our study, we expect that the coupled concept should be applied to other hydrogenation reactions with larger market sizes (e.g., acetone to isopropanol, CO₂ to hydrocarbons).

2) Is the LCA analysis taking into account the device footprint/area preparation/installation?

Response: We appreciate the suggestion of the reviewer. We have conducted a preliminary system-level analysis of our coupled device, which has considered these additional factors., Please refer to our response to **comment #7 of Reviewer 1**.

3) What about the LCA of itaconic acid manufacturing, which is used as a feedstock in this process?

Response: We thank the reviewer for bringing up this point. Indeed, itaconic acid is used as a feedstock in this process and the LCA of itaconic acid was used to determine its primary energy demand.

Associated changes to the manuscript:

- Page 27: "Nieder-Heitmann et al. has reported an LCA study of plant-scale mass production of IA, which documented all the materials and emissions flow.⁵⁶"

4) Is the energy demand for IA, MSA, and homogeneous catalyst separation and purification considered?

Response: This is indeed an interesting and very relevant question. We did not initially include it for the device-level analysis. However, we have now performed a preliminary system-level analysis, in which the energy demand for separation and purification is considered. Please refer to our response to **comment #10 of Reviewer 1**.

5) On line 281, the authors claimed that preliminary H₂-to-MSA conversion studies show conversion efficiency of up to 60%; may this preliminary data be included in the manuscript? This real data will strengthen the manuscript.

Response: We agree with the reviewer that including the experimental data would support and strengthen our analysis. However, the data has been included in a manuscript that reports the demonstration of the device, and the manuscript is currently under review in another journal. Figure R1 below shows the amount of produced MSA and the associated H₂-to-MSA conversion from our demonstrated coupled photoelectrochemical hydrogenation device. We sincerely hope that the experimental manuscript will also be published and available soon.

Figure R1. The amount of produced MSA generated using (a) coupled electrochemical hydrogenation of IA using dimensionally stable anode (DSA®) and Pt-cathode and (b) coupled photoelectrochemical hydrogenation of IA using BiVO₄ photoanode and Pt cathode. The H₂-to-MSA conversion values obtained in both experiments are shown.

6) Why was the Rh-TPPTS catalyst selected for this scheme? Do the reaction conditions match well with the PEC device's working conditions? Could this hydrogenation process be done electrochemically by the electrocatalyst at the dark cathode?

Response: Rh-TPPTS catalyst was selected since hydrogenation of IA to MSA using this catalyst has been reported to proceed at room temperature and atmospheric pressure, which is the condition typically used in PEC H₂ production (see Milano-Brusco, Schwarze, et al. *Cat. Ind. & Eng Chem Res.* 47, 2008, 7586). The hydrogenation process can also be performed electrochemically by a heterogeneous electrocatalyst using a dark cathode (see Holzhäuser et al., *Green. Chem.* 19, 2017, 2390), but in our experimental study, we show that the coupled approach is advantageous from the stability point-of-

view. As shown in Fig. R2 below (*only shown for reviewers here since the manuscript that contains the data is under review at another journal*), the non-coupled electrochemical hydrogenation (i.e., directly on the surface of the electrode) initially shows the production of MSA, but the production rate decreases with time and the generation of MSA is terminated after ~120 min. In contrast, the coupled electrochemical hydrogenation device shows a continuous production of MSA. More importantly, our coupled approach is also more attractive from the flexibility point-of-view. By changing the concentration or even the types of feedstocks and homogenous catalysts, we can control the hydrogenation rate and access other hydrogenation reactions on demand with the same device without the need to change any components. We have now added this information and arguments to the revised manuscript.

Figure R2. The amount of produced MSA generated using Pt electrode under (a) direct electrochemical hydrogenation of IA (i.e., no homogenous catalyst) and (b) coupled electrochemical hydrogenation of IA (i.e., with homogenous catalyst).

Associated changes to the manuscript:

- Page 4: "This specific catalyst is considered here since it has been reported to operate at room temperature and atmospheric pressure,²² which is the condition commonly used in PEC H₂ production."
- Page 18 and 19: "Although some of these hydrogenation reactions may also be performed through electrochemical processes directly on suitable dark electrodes,^{39,40} the coupled concept proposed here introduces an important flexibility advantage in that the hydrogenation rate and the type of hydrogenation reactions performed can be controlled on demand using the same device without the need to change any components by simply changing the concentration and/or the types of feedstocks and homogenous catalysts."

7) Is this co-evolved H₂ and MSA process in a PEC device more favorable than the PV-electrolysis scheme?

Response: We thank the reviewer for suggesting us to make a performance comparison between the PV-electrolysis scheme and our study. The cumulative energy demand value for H₂ from PV-electrolysis has been added to the manuscript and **Table S5** for comparison.

Associated changes to the manuscript:

- Page 10: "The production of hydrogen using electricity from a photovoltaic system to drive a water electrolysis unit has been reported to have an exergy efficiency of 0.64,²⁸ which translates to a CED value of 187.5 MJ/kg H₂."
- Supplementary information sheet, Page 8:

Table S5. Cumulative energy demand (CED) for H₂ production with different methods.

Technology	CED (MJ/kg H ₂)
Wind/electrolysis ¹⁹	9.1
Natural gas steam reforming ²⁰	183.2
PV-electrolysis ²¹	187.5
Country grid mix/electrolysis ²²	192 - 217.8
PEC water splitting (microwire Si) ¹⁸	10 - 194
PEC water splitting (SHJ/BiVO ₄ ; this study)	34 - 680

REVIEWERS' COMMENTS

Reviewer #1 (Remarks to the Author):

All my comments have been adequately addressed. The authors responded well to the two other reviewer's comments. I believe the paper is now suitable for publication.

Reviewer #2 (Remarks to the Author):

I am satisfied with the response of the authors to all the queries/suggestions I made in my first review.

I also found it very interesting to read the author responses to the comments of the other reviewers and I think the paper is now in better shape and merits publication.

Reviewer #3 (Remarks to the Author):

Dr. Adbi and colleagues thoroughly revised their manuscript in response to all the reviewers' comments. They conducted additional analyses on the energy requirements of additional components such as the IA feedstock, product separation and catalyst separation, land area, and other H₂ production methods. They made every effort to include error bars to demonstrate the range of uncertainties in their analyses. Because PEC and homogeneous IA hydrogenation are both in their early stages, the uncertainties in their analysis are unexpectedly high.

While the authors claim that the IA-to-MSA conversion is only a model system and that the concept should be applicable to other hydrogenation reactions with a larger market size, it is unclear whether this PEC device concept would be compatible because most hydrogenation reactions are performed at higher pressures, higher temperatures, and with heterogeneous catalysts to achieve full conversion and the highest production rate to ensure easy product separation. These industrially relevant conditions contradict the conditions proposed in this manuscript. Nevertheless, the concept presented here is intriguing and suggests that there may be a new way to apply valuable PEC knowledge for alternative chemical production.

Manuscript ID: NCOMMS-22-33098A

Title: Sustainable H₂ production and hydrogenation of chemicals in a coupled photoelectrochemical device – a life cycle net energy assessment

REVIEWER REPORT(S): (Authors' response and associated changes in blue)

Reviewer #1 (Remarks to the Author):

All my comments have been adequately addressed. The authors responded well to the two other reviewer's comments. I believe the paper is now suitable for publication.

Response: We are pleased to know our revision has adequately addressed the reviewer's comments. We thank the reviewer for considering our paper to be suitable for publication.

Reviewer #2 (Remarks to the Author):

I am satisfied with the response of the authors to all the queries/suggestions I made in my first review. I also found it very interesting to read the author responses to the comments of the other reviewers and I think the paper is now in better shape and merits publication.

Response: We are happy that the reviewer is satisfied with our responses. We thank the reviewer for the initial constructive comments and their support for the publication of our paper.

Reviewer #3 (Remarks to the Author):

Dr. Abdi and colleagues thoroughly revised their manuscript in response to all the reviewers' comments. They conducted additional analyses on the energy requirements of additional components such as the IA feedstock, product separation and catalyst separation, land area, and other H₂ production methods. They made every effort to include error bars to demonstrate the range of uncertainties in their analyses. Because PEC and homogeneous IA hydrogenation are both in their early stages, the uncertainties in their analysis are unexpectedly high.

While the authors claim that the IA-to-MSA conversion is only a model system and that the concept should be applicable to other hydrogenation reactions with a larger market size, it is unclear whether this PEC device concept would be compatible because most hydrogenation reactions are performed at higher pressures, higher temperatures, and with heterogeneous catalysts to achieve full conversion and the highest production rate to ensure easy product separation. These industrially relevant conditions contradict the conditions proposed in this manuscript. Nevertheless, the concept presented here is intriguing and suggests that there may be a new way to apply valuable PEC knowledge for alternative chemical production.

Response: We thank the reviewer for acknowledging our thorough revision in response to all the reviewers' comments. We also agree with the reviewer's point that there might be compatibility issues between the conditions used in many of the current hydrogenation reactions and the conditions in the PEC device proposed in our study. We have therefore modified our claim by adding a boundary condition that the feasible hydrogenation reactions are those possible to be performed at e.g., near-atmospheric pressure and ambient temperature. We briefly note, however, that PEC devices operating at elevated pressure and temperature are not impossible to achieve, since vapor-based PEC devices have

actually been demonstrated in the literature (e.g., ref 31 in our paper and Kistler et al. J. Electrochem. Soc. 166, 2019, H3020).

Associated changes to the manuscript:

- Page 13, lines 304-306: *"Finally, other hydrogenation reactions possible to be performed at conditions relevant to that of PEC devices (e.g., near-atmospheric pressure, ambient temperature), such as levulinic acid to γ -valerolactonate³⁷ or acetone to isopropanol,³⁸ need to be explored to determine the optimum reaction and/or identify the general applicability of the coupled device concept."*